# AFM imaging reveals the unreconstructed α-Al₂O₃(0001) surface to be inhomogeneous and rough

Johanna I. Hütner-Reisch ®[1,2], Andrea Conti ®[1,2], David Kugler ®[1], Florian Mittendorfer ®[1], Michael Schmid ®[1], Ulrike Diebold ®[1] & Jan Balajka ®[1] ✉

Alumina (Al₂O₃) is a key material for thin-film growth and heterogeneous catalysis, where the atomic surface structure critically impacts performance. Using noncontact atomic force microscopy (nc-AFM) combined with density functional theory (DFT) calculations, we challenge the common assumption that the unreconstructed α-Al₂O₃(0001) surface is atomically flat and uniformly Al-terminated. This widely accepted bulk termination satisfies polarity compensation requirements but results in highly undercoordinated Al cations at the surface. Despite substantial inward relaxation of these Al cations, we find that the (1 × 1) surface remains inherently metastable, relative to the thermodynamically stable $(\sqrt{31} \times \sqrt{31})\,R \pm 9°$ surface reconstruction that forms at high temperatures above 1000 °C. Nc-AFM imaging of the unreconstructed surface reveals a rough and disordered morphology, with only nanometer-scale regions exhibiting the ordered Al-terminated (1 × 1) structure. Our results show that the unreconstructed Al₂O₃(0001) surface is intrinsically inhomogeneous, reconciling conflicting experimental observations and challenging the validity of commonly used atomistic models.

The thermodynamically stable form of the α-Al₂O₃(0001) surface is the stoichiometric $(\sqrt{31} \times \sqrt{31})\,R \pm 9°$ reconstruction[1], which forms at high temperatures above 1000 °C. Under conditions relevant to technological applications, the surface typically remains unreconstructed. In the corundum lattice of Al₂O₃, a non-polar (Tasker-type II[2]) (0001) surface can only be created by cutting between aluminum planes (Fig. 1a), exposing threefold-coordinated Al cations at the surface (Fig. 1b, Supplementary Data 1). This highly undercoordinated termination results in a high surface energy of the bulk-truncated surface. To increase their coordination, the surface Al atoms relax strongly inward into the underlying oxygen plane (Fig. 1c, Supplementary Data 2). Our DFT calculations show an inward displacement of the surface Al atoms by ≈ 0.7 Å, in agreement with earlier theoretical works[3–5] and consistent with studies of the unreconstructed surface by diffraction and scattering experiments[6–10]. Although the surface relaxation lowers the energy by more than 50% (blue dotted line in

Fig. 1d), it does not fully stabilize the surface. Thermodynamic stability is achieved only upon formation of the $(\sqrt{31} \times \sqrt{31})\,R \pm 9°$ reconstruction, which eliminates the undercoordinated surface Al sites via bonding to subsurface oxygen[1], leading to a further ≈ 30% reduction in surface energy (red solid lines in Fig. 1d). This reconstruction, however, only forms at high temperatures, when the increased atomic mobility enables an extensive reorganization across multiple atomic layers. The reconstruction preserves the overall stoichiometry, i.e., the composition of both the unreconstructed and the reconstructed surfaces remains Al₂O₃. The bulk-terminated (1 × 1) surface remains energetically unfavorable across the entire range of oxygen chemical potentials (see Computational Methods for conversion to O₂ pressures and temperatures). This Al-terminated (1 × 1) surface (Fig. 1c,e) is therefore an intrinsically metastable, transient structure of the Al₂O₃(0001) surface. Yet, it has long served as the standard model for epitaxial growth, adsorption, and surface chemistry studies[7,11–26].

[1]Institute of Applied Physics, TU Wien, Vienna, Austria. [2]These authors contributed equally: Johanna I. Hütner-Reisch, Andrea Conti.
✉e-mail: jan.balajka@tuwien.ac.at

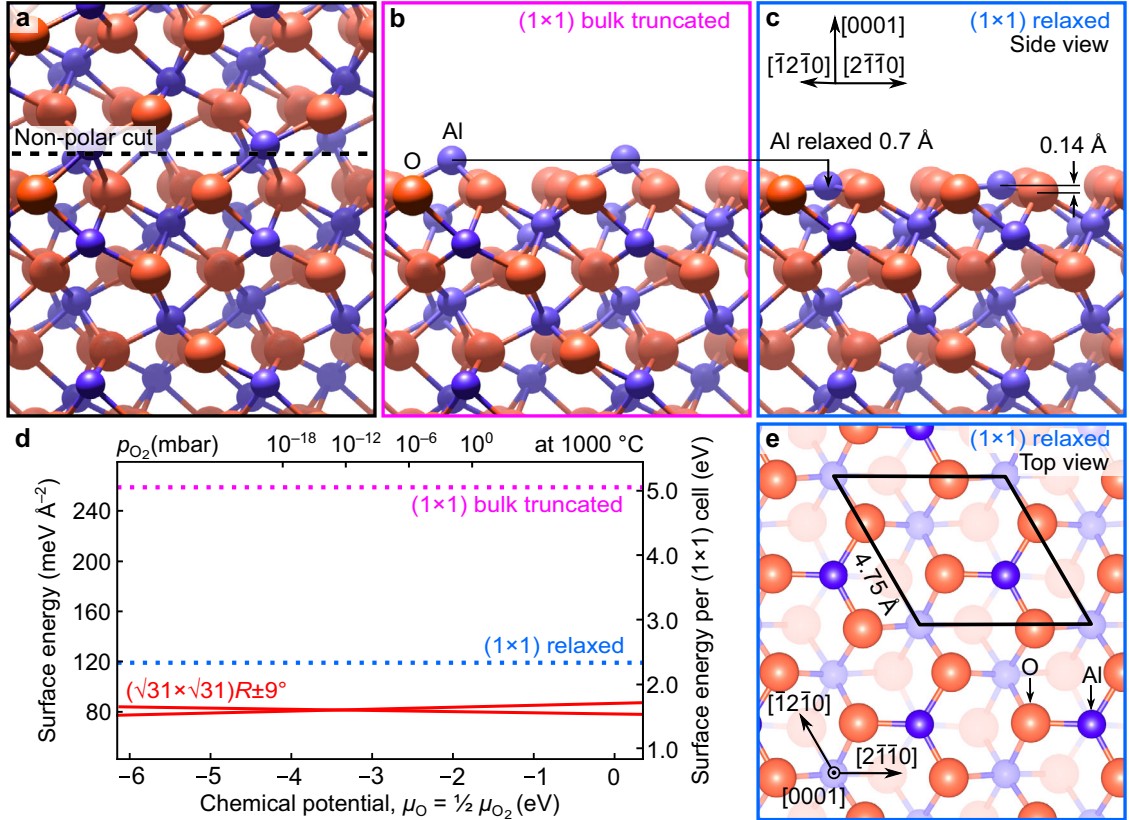

**Fig. 1 | The Al-terminated Al$_2$O$_3$(0001)-(1 × 1) surface is metastable. a** Corundum α-Al$_2$O$_3$ bulk structure with the non-polar (0001) cutting plane (dashed line). **b** Bulk-truncated surface exposing undercoordinated Al cations (Supplementary Data 1). **c, e** DFT-relaxed Al-terminated (1 × 1) surface: surface Al atoms relax inward into the O plane, adopting nearly threefold-planar coordination (Supplementary Data 2). The (1 × 1) unit cell is marked by a black rhombus. **d** Ab initio phase diagram: although the relaxation lowers surface energy, the (1 × 1) termination remains unfavorable relative to the stable ($\sqrt{31} \times \sqrt{31}$) $R \pm 9°$ reconstruction. Red lines indicate the coexisting O-rich and O-poor variants of the reconstructed surface[1]. (a,c) Adapted from ref. 1 with permission (Copyright 2024, AAAS).

Experimental results, however, suggest a more complex reality. A central enigma is the reactivity of the unreconstructed surface toward water. Theory consistently predicts facile H$_2$O dissociation with low activation barriers at the Al-terminated (1 × 1) surface[16,17,27–31], in line with the expectation that undercoordinated Al cations should be highly reactive. Thermal desorption and vibrational spectroscopy experiments, however, have yielded contradictory results. While some studies report dissociative adsorption[19,32–34], others find that the unreconstructed surface is not easily hydroxylated, or even unreactive under ambient conditions[20,21,28]. Under ultrahigh vacuum (UHV) conditions, hydroxylation is typically limited to low coverages[34–36], which was interpreted as dissociation at defects, steps and other surface heterogeneities[22,32,35,36]. Full coverage has only been achieved at elevated pressures[34], unusually high exposures[35,36], or with supersonic molecular beams[28]. Water desorption temperatures vary by several hundred Kelvin, and the interpretation of desorption spectra remains controversial. Petrik et al.[20] found only molecular adsorption by infrared spectroscopy despite observing thermal desorption peak shapes resembling those previously attributed to dissociatively adsorbed water[35,36]. Thissen et al. proposed that molecular and dissociative species coexist[22]. Here, we show that the disagreement between theory and experiment arises from the fact that the majority of the unreconstructed Al$_2$O$_3$(0001) surface differs from the commonly assumed idealized structure model.

Under wet conditions, the surface transforms into a gibbsite-like Al(OH)$_3$ termination that is O-terminated, fully hydroxylated, and (1 × 1) ordered[7,15,37]. Differences in preparation conditions may lead to different surface terminations[28], as reflected by the wide range of

reported isoelectric points (pH 3.1–8) for nominally identical single crystals[38]. These ambiguities highlight the need for a well-defined model surface with known surface structure as a prerequisite for understanding and controlling alumina surface chemistry at the molecular level.

Spatially resolved imaging offers a route to visualize the surface structure and resolve the discrepancies between the predicted and observed reactivity of alumina surfaces. Yet atomic resolution on the clean, unreconstructed Al$_2$O$_3$(0001) surface has remained elusive[39,40]. Recent advances in noncontact atomic force microscopy (nc-AFM), particularly the development of the qPlus sensor[41], have enabled major progress in atomic-scale characterization of insulating materials, and judicious tip functionalization can now provide chemical sensitivity on the single-atom level[42]. Applied to the ($\sqrt{31} \times \sqrt{31}$) $R \pm 9°$ reconstruction, this methodology has already provided a detailed atomic model of this complex, lowest-energy structure[1].

In this work, we apply qPlus-based nc-AFM to the unreconstructed Al$_2$O$_3$(0001) surface and show that, contrary to the prevailing view, it is intrinsically inhomogeneous and rough, with the ordered Al-terminated (1 × 1) structure occurring only as a minority termination.

## Results

### Rough morphology of unreconstructed Al$_2$O$_3$(0001)

Al$_2$O$_3$(0001) samples annealed in a tube furnace in air exhibited a well-defined terrace-step morphology in ambient AFM, albeit with noticeable roughness within individual terraces (Supplementary Fig. 1). After transfer to UHV, the samples were further annealed up to 900 °C in low partial O$_2$ pressure up to 10$^{-6}$ mbar, below the temperature where the ($\sqrt{31} \times \sqrt{31}$) $R \pm 9°$ reconstruction forms, as confirmed by the absence

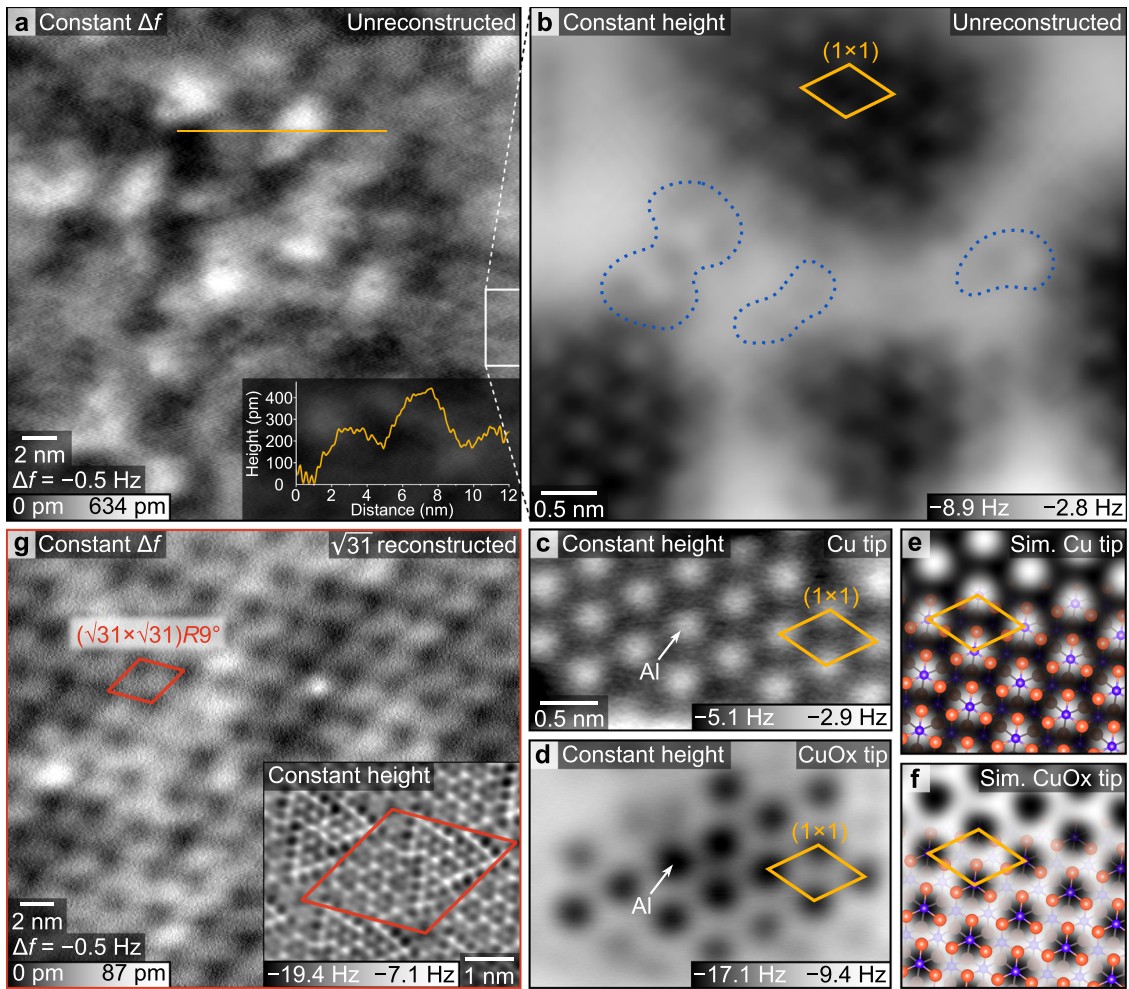

**Fig. 2 | Morphology and atomic structure of Al₂O₃(0001). a** Nc-AFM topography of the unreconstructed surface obtained by annealing at ≈ 820 °C, recorded with a Cu-terminated tip in constant-frequency-shift mode ($\Delta f = -0.5$ Hz). Bright areas correspond to topographic protrusions and dark areas to depressions; the overall height range spans approximately three atomic steps (interlayer distance 216 pm). A representative height profile (along the orange line) is shown in the inset. **b** Constant-height image acquired with a Cu-terminated tip showing nanometer-sized (1 × 1) islands surrounded by lower-lying and disordered regions. Dark corresponds to attractive tip-surface interaction and indicates topographically higher islands, brighter areas represent lower-lying regions. Faint protrusions in the lower-lying areas (blue dotted lines) indicate the absence of periodic order. The frame at the right of the topography image (**a**) indicates the position of the constant-height image in (**b**). **c, d** High-resolution nc-AFM images of (1 × 1) islands acquired with (**c**) a Cu tip and (**d**) a CuOx tip, showing inverted contrast at surface Al sites. The measured size of the (1 × 1) unit cell (orange) is consistent with the lattice constant of 475 pm. **e, f** Simulated AFM images reproduce the contrast of the Al-terminated (1 × 1) surface for (**e**) a positively charged Cu tip and (**f**) a negatively charged CuOx tip. **g** Nc-AFM topography of the ($\sqrt{31} \times \sqrt{31}$) $R \pm 9°$ reconstructed surface (annealed at 1300 °C) with a height variation below a single atomic step; the inset shows atomic-scale contrast within the reconstruction unit cell (outlined in red). Imaging and simulation parameters and details of sample preparation are provided in the Methods section and Supplementary Note 1.

of reconstruction spots in low-energy electron diffraction (LEED) (Supplementary Fig. 2). Similar preparation procedures have been employed in the literature to obtain the unreconstructed Al₂O₃(0001) surface, although the reported annealing temperatures vary[6,9,10,43–45]. High-temperature annealing removed surface contamination, as verified by X-ray photoelectron spectroscopy (XPS) (Supplementary Fig. 3). No contaminants were detected, aside from a minute trace of fluorine on the sample shown in Fig. 2. This fluorine signal was not reproduced on other samples exhibiting the same morphology and does not affect the results presented.

In UHV, the morphology was examined with nc-AFM in constant-frequency-shift mode, where the tip follows the surface at a distance dominated by long-range forces. The unreconstructed surface appeared rough, with nanoscale height variations spanning several monoatomic steps (Fig. 2a) and no periodic order. In contrast, the ($\sqrt{31} \times \sqrt{31}$) $R \pm 9°$ reconstructed surface was markedly smoother, with height variations well below a single atomic step (Fig. 2g), arising mainly from corrugation within the reconstruction unit cell (Fig. 2g, inset) and clearly apparent periodic order.

## Atomic-scale structure of Al-terminated (1 × 1) domains

In regions with low corrugation, atomic resolution was achieved in constant-height mode (Fig. 2b–d), at a closer imaging distance. The images confirmed the rough and laterally inhomogeneous morphology of the unreconstructed Al₂O₃(0001) surface, with nanometer-sized islands of ordered hexagonal symmetry consistent with a (1 × 1) unit cell (Fig. 2b). The chemical identity of these locally Al-terminated regions was established using functionalized tips: with a positively charged Cu tip[46,47], Al cations appeared as bright (repulsive) features (Fig. 2c), whereas with a negatively charged CuOx-functionalized tip[42,48] the same sites appeared dark (attractive) (Fig. 2d). The termination of the CuOx tip was verified by a "fingerprint" image shown in Supplementary Fig. 4. The contrast inversion between the differently terminated tips, closely reproduced by AFM simulations (Fig. 2e,f),

arises from electrostatic interaction between the tip and the surface Al atoms. The same qualitative contrast was consistently observed over a range of tip–sample distances (Supplementary Fig. 5) and is well described by the calculated electrostatic potential near the surface (Supplementary Fig. 6). Approaching the tip closer to the surface led to increased interaction and imaging instabilities before entering the Pauli repulsion regime. The images were thus acquired at tip–sample distances dominated by the electrostatic interaction to ensure stable imaging conditions. In both experiment and simulation, the contrast is dominated by the undercoordinated surface Al cations, whereas the close-packed oxygen sublattice is not resolved. Outside the $(1 \times 1)$ ordered islands, the surface appeared disordered; faint protrusions (highlighted in Fig. 2b) indicate the absence of a periodic structure just below the Al-terminated islands.

### Stoichiometric and predominantly oxygen-terminated surface

XPS spectra acquired at different emission angles (Supplementary Fig. 7) enable a qualitative comparison of surface and bulk composition. The Al $2p$ intensity measured at 70° emission (enhanced surface sensitivity) shows a markedly lower aluminum-to-oxygen ratio compared to the spectrum collected in normal emission, indicating a lower surface concentration of aluminum on the unreconstructed surface. This observation is incompatible with a uniformly Al-terminated surface and supports our nc-AFM finding that the Al-terminated $(1 \times 1)$ structure occurs only as a minority phase, contributing minimally to the overall XPS signal. Comparison with the $(\sqrt{31} \times \sqrt{31}) R \pm 9°$ reconstructed surface (Supplementary Fig. 8) shows a nearly unchanged [O]/[Al] ratio, consistent with the previous finding that wide–band-gap insulators are difficult to reduce[1]. The O $1s$ peak of the reconstructed surface is considerably broader than that of the unreconstructed surface, reflecting the diverse coordination environments present within the large $\sqrt{31}$ unit cell.

### Surface flattening into a stable $\sqrt{31}$ reconstruction

Even after stepwise annealing in small temperature increments, the unreconstructed surface did not evolve into a flat, laterally uniform morphology. The $(1 \times 1)$ ordered islands did not grow laterally with increasing temperature. Instead, their size increased only at the onset of surface reconstruction when the islands adopted structural motifs characteristic of the $(\sqrt{31} \times \sqrt{31}) R \pm 9°$ reconstruction (Supplementary Fig. 9). Above ≈ 1000 °C, the surface transformed into an atomically flat and well-ordered $(\sqrt{31} \times \sqrt{31}) R \pm 9°$ reconstruction (Fig. 2g). No intermediate reconstructions were observed on clean samples, whose surface cleanliness was verified by XPS. LEED patterns recorded after successive annealing steps (Supplementary Fig. 2) confirm this direct transition, in contrast to earlier reports of intermediate reconstructions[49].

Once formed, the reconstructed surface remained stable and did not revert to the $(1 \times 1)$ structure, even after annealing in an $O_2$ background up to $10^{-4}$ mbar (Supplementary Fig. 10a). This observation contrasts with earlier reports suggesting reversibility under oxidizing conditions[50], but agrees with recent results by Smink et al.[51]. The ab initio phase diagram (Fig. 1d) further supports this irreversibility, showing that the reconstructed phase is thermodynamically favored across the entire range of accessible oxygen chemical potentials. The $(\sqrt{31} \times \sqrt{31}) R \pm 9°$ reconstruction also exhibited environmental stability, remaining unchanged after controlled exposure to ultrapure liquid water (Supplementary Fig. 10b).

## Discussion

These results demonstrate that the Al-terminated $(1 \times 1)$ structure is not a realistic description of the unreconstructed $Al_2O_3(0001)$ surface. Instead, the surface is rough and inhomogeneous, with only nanometer-scale islands of ordered $(1 \times 1)$ termination. Reconciling this result with available structural measurements requires a closer look at the original data. Quantitative LEED-IV measurements on insulating samples are intrinsically challenging because charging restricts the electron-energy range, and inaccurate electron energies due to charging compromise the $R$-factor, the accepted measure of agreement between theory and experiment. References[8,9] reported satisfactory $R$-factors only when introducing large anharmonicity of the top Al atoms, an effect that, as noted in ref. 9, could equally indicate static disorder. Grazing-incidence X-ray diffraction measurements often cited as evidence for Al termination[10] tested different structure models that yielded similar $\chi^2$ values. The same method also resulted in an early model of the $(\sqrt{31} \times \sqrt{31}) R \pm 9°$ surface[52], which has since been substantially revised based on recent nc-AFM measurements[1]. LEED patterns obtained in the present work (Supplementary Fig. 2) are comparable in quality to those previously reported[35,36,49,50], suggesting that the observed $(1 \times 1)$ periodicity in electron diffraction primarily reflects the underlying bulk lattice, with a thin, disordered surface layer covering most regions of the sample. In comparison with the $(\sqrt{31} \times \sqrt{31}) R \pm 9°$ reconstruction, the $(1 \times 1)$ LEED pattern of the unreconstructed surface exhibits weaker spot intensities and enhanced diffuse background (Supplementary Fig. 11). Consistent with this observation, RHEED data reported in the literature show broader diffraction spots and stronger diffuse background intensity in the $(1 \times 1)$ patterns of the unreconstructed $Al_2O_3(0001)$ surface compared to those of the $(\sqrt{31} \times \sqrt{31}) R \pm 9°$ reconstruction[51,53,54].

The rough and irregular surface observed with nc-AFM helps reconcile prior conflicting reports of water adsorption on $Al_2O_3(0001)$. The images reveal only nanometer-scale islands with the Al-terminated $(1 \times 1)$ structure, while the surrounding regions appear disordered and remain inaccessible to atomic-resolution imaging. Considering the inertness of amorphous alumina layers formed upon oxidation of metallic aluminum, these disordered regions likely contain configurations in which surface atoms adopt higher coordination and are therefore unreactive[55]. Such regions would only allow molecular water adsorption, whereas the small degree of dissociative adsorption, previously attributed to surface defects, can be rationalized as occurring at the $(1 \times 1)$ Al-terminated patches, which cover only a small fraction of the surface.

The lateral inhomogeneity of the unreconstructed $Al_2O_3(0001)$ surface has important implications for epitaxial growth on sapphire. The nanometer-scale $(1 \times 1)$ domains separated by disordered regions provide a structurally non-uniform template and may influence film quality, particularly during the early stages of growth. For two-dimensional materials and ultrathin films, where the interface structure plays a critical role, such irregularities may strongly affect nucleation and growth. In contrast, the thermodynamically stable $(\sqrt{31} \times \sqrt{31}) R \pm 9°$ reconstruction yields a flat and well-ordered morphology and has been reported to promote adhesion during metal growth[43] and recently shown to substantially improve the quality of $WS_2$ grown on sapphire[56].

At this point, it remains speculative why the unreconstructed surface is rough and whether this is an intrinsic property. Even for a simplified model system (see Computational Methods, Supplementary Fig. 12 and Supplementary Data 3), the energetic cost of forming steps on unreconstructed $Al_2O_3(0001)$ is low, which may facilitate roughening and contribute to the observed morphology. As the set of structural models explored is not exhaustive, configurations with even lower step energies may exist. The stepped configurations allow locally higher coordination of surface Al atoms compared to a flat, Al-terminated $(1 \times 1)$ structure, thereby reducing the energetic penalty associated with undercoordinated Al sites. In addition to low step energies, the observed roughness of the unreconstructed surface may also originate from the surface termination present before UHV preparation. Samples exposed to ambient conditions can develop a gibbsite-like $Al(OH)_3$ termination (a fully hydroxylated O-termination)[57]. An Al-terminated

surface and a hydroxylated, O-terminated surface should be interconvertible by de/hydration, as shown in the molecular dynamics study of Hass et al.[15]. However, this transformation is hindered by substantial kinetic barriers and does not occur readily, as demonstrated by Yue[28] and others[20,21,29,34–36,58]. Converting a hydroxylated surface into an Al-terminated structure requires substantial mass transport, inherently leading to roughness (see Supplementary Fig. 13 and Supplementary Data 4–6).

In summary, these results show that the simple, Al-terminated (1 × 1) surface, widely used in theoretical models of unreconstructed $Al_2O_3(0001)$, is not observed in experiment. Even under pristine UHV preparation conditions, the unreconstructed surface is rough and inhomogeneous, and the expected (1 × 1) Al-terminated surface occurs only as a minority phase on a surface that nonetheless exhibits a (1 × 1) periodicity in electron diffraction. Recognizing this intrinsic complexity calls for a re-evaluation of how alumina is used as a substrate for thin-film growth and how its surface chemistry is understood.

## Methods

### Sample preparation
Polished α-$Al_2O_3(0001)$ single crystals from Crystec GmbH were cleaned by several sonication cycles in a heated (40 °C) pH-neutral detergent solution (3% Extran MA 02, Merck) followed by thorough rinsing with ultrapure $H_2O$ (Milli-Q, Millipore, 18.2 MΩ cm, <3 ppb total organic carbon) until no polishing residues were observed by ambient AFM (Agilent 5500). The cleaned crystals were then annealed in air at 1100 °C for 10 h in a tube furnace, resulting in wide atomic terraces (of ≈ 500 nm) separated by straight, monoatomic step edges ≈ 220 pm high (Supplementary Fig. 1). Samples were mounted on tantalum Omicron-type sample plates using spot-welded Ta wires. To minimize contamination, the sample plates and wires, as well as the crystals were each separately boiled in ultrapure water, followed by an additional boiling after assembly. The samples were then introduced into a UHV system and annealed at 820 to 900 °C in 1 × 10⁻⁶ mbar $O_2$ for 1.5 h in the preparation chamber (base pressure <2 × 10⁻¹⁰ mbar) using a tungsten filament heater, calibrated by measuring the temperature of a blank sample plate with a thermocouple vs. filament power. This annealing step leads to desorption of impurities. Sample cleanliness was checked by XPS, see Supplementary Fig. 3. To prevent adsorption of residual gases, the annealed samples were transferred to the analysis chamber (base pressure <1 × 10⁻¹¹ mbar) at elevated temperature, prior to cooling to room temperature.

### Noncontact atomic force microscopy
The nc-AFM measurements were performed at 4.7 K using an Omicron qPlus LT STM/AFM equipped with a cryogenic differential amplifier[59]. A qPlus sensor[41] ($f_0$ = 30.8 kHz, $k$ = 1800 N m⁻¹, $Q$ = 11,000–16,000) with an electrochemically etched tungsten tip was used for imaging. The AFM tips were conditioned in UHV by field emission, Ar⁺ self-sputtering[60], and voltage pulsing on clean Au(100) and Cu(110) surfaces. Oxygen-terminated tips were formed by controlled indentation and voltage pulsing on a Cu(110) surface partially covered with an oxygen-induced added-row reconstruction, prepared by brief (15 s) exposure to $O_2$ (2 × 10⁻⁸ mbar) at 250 °C. A stable CuOx tip termination was confirmed by evaluating contrast variations over the Cu-O rows, as described in ref. 42; see Supplementary Fig. 4. Unless otherwise noted, imaging was conducted in constant-height mode using sample bias voltages between 0.0 V and −0.2 V to minimize the local contact potential difference (LCPD). The images were filtered in the Fourier domain to remove electrical and mechanical noise.

### X-ray photoelectron spectroscopy
XPS measurements were conducted in the preparation chamber using a SPECS XR 50 non-monochromatized Al $K_α$ source and a SPECS Phoibos 100 hemispherical analyzer. Spectra were acquired at pass energies of 20 eV (individual spectra) and 60 eV (survey scans) in normal emission or at an emission angle of 70° relative to the surface normal to enhance surface sensitivity. To compensate for surface charging, a uniform energy offset of 7.8 eV was applied to align the O 1$s$ binding energy to a nominal reference value of 531.0 eV. The L Shirley (linear+Shirley) background of CasaXPS (version 2.3.25) was subtracted from the O 1$s$ and Al 2$p$ spectra before analysis. For visual comparison of peak shapes and relative intensities, each spectrum was normalized to the integrated area of the corresponding O 1$s$ peak. The Al 2$p$ spectra were divided by the same normalization factor, enabling comparison of relative Al contributions between different emission angles and surface preparations. The relative concentrations of O and Al were estimated using atomic sensitivity factors[61] of 0.711 for O 1$s$ and 0.234 for Al 2$p$ for the background-subtracted peak areas.

### Low-energy electron diffraction
LEED patterns were recorded in the same chamber using a SPECS ErLEED 150. Screen inhomogeneities were corrected using flat-field (LEED image of a polycrystalline holder) and dark-frame (screen voltage off) images[62].

### Ab initio calculations
DFT calculations were performed with the r²SCAN meta-GGA (generalized gradient approximation) exchange-correlation functional[63], as implemented in the Vienna Ab-initio Simulation Package (VASP)[64,65]. The lattice parameters $a$ and $c$ of the α-$Al_2O_3$ bulk cell deviated from experimental values[66] only by +0.1% and −0.2%, respectively, indicating that the bulk structure is reproduced well by this functional. The bulk unit cell was optimized using a cutoff energy of 800 eV, and a 3 × 3 × 1 Monkhorst-Pack grid to integrate the Brillouin zone of the $Al_2O_3$ hexagonal cell. Slab models were created from the DFT-optimized bulk structure by cleaving along the non-polar (0001) planes (Fig. 1a). Periodically repeated stoichiometric slabs with two equivalent faces were decoupled by a vacuum region of 15 Å. The surface calculations were performed using a lower cutoff energy of 500 eV and the same k-point mesh sampling used for the bulk cell. Structures were optimized using the conjugate-gradient algorithm until the norms of all the forces on the atoms were smaller than 0.01 eV Å⁻¹.

To compare surfaces with different stoichiometries, the surface free energies were evaluated using the formalism of ab initio atomistic thermodynamics[67]. Zero-point energy (ZPE) and vibrational entropy contributions were not explicitly calculated, as these terms largely cancel out when comparing related oxide terminations and therefore do not alter the qualitative stability ordering. The oxygen chemical potential $\mu_O(T, p)$ in the ab initio phase diagram in Fig. 1d is defined as:

$$\mu_O(T,p) = \frac{1}{2}\mu_{O_2}(T,p) = \mu_O(T,p^o) + \frac{1}{2}kT\ln\left(\frac{p}{p^o}\right),$$

where $\mu_O(T,p^o)$ describes the temperature dependence at a reference pressure $p^o$ (ref. 67). The energy of an isolated $O_2$ molecule in the gas phase was obtained in a spin-polarized calculation. This calculation converged to a magnetic moment of 2.0 $\mu_B$, consistent with the triplet ground state of molecular oxygen. The r²SCAN functional was employed because it significantly improves predicted heats of formation for oxides[68] and provides a more accurate description of $O_2$ binding energies and magnetic states compared to previous GGA functionals[63], ensuring reliable reference energies.

### Computational modeling of surface steps
Surface steps on α-$Al_2O_3(0001)$ were modeled using a (2 × 6) surface unit cell (in-plane dimensions 9.51 Å × 28.53 Å), derived from the relaxed (1 × 1) slab model (Fig. 1c). To investigate steps of varying depth and width, between one and twelve $Al_2O_3$ units were systematically

removed from the topmost layers, while maintaining the overall stoichiometry. These asymmetric slabs were terminated by a bulk-truncated Al layer at the bottom to avoid a polar surface. During geometry optimization, the bottom-most layers of the slab were fixed, while the upper layers were allowed to relax. Periodically repeated slabs were separated by a 15 Å vacuum gap.

To efficiently explore the large configuration space, machine-learned force fields (MLFFs) based on the Gaussian Approximation Potential (GAP) approach (VASP version 6.5.1) were employed. The MLFFs were trained sequentially on several stoichiometric systems: bulk $\alpha$-$Al_2O_3$, $(1 \times 1)$ and $(2 \times 2)$ slabs, $(2 \times 4)$ slabs containing various step structures, a three-dimensional cluster derived from bulk $\alpha$-$Al_2O_3$, and $(2 \times 6)$ stepped slabs. The training data were generated from molecular dynamics (MD) simulations performed between 1000 to 2300 K (near the melting point), resulting in a robust MLFF based on more than 2300 structures.

The trained MLFF was then used to identify low-energy step configurations via parallel tempering simulations[69], followed by geometry relaxations using simulated annealing. These simulations were carried out in prediction-only mode, without additional ab initio calculations. The parallel tempering simulations employed 24 replicas with logarithmically spaced temperatures from 150 to 2400 K, ensuring high exchange probabilities. Swaps between adjacent replicas were attempted every 200 steps, yielding acceptance ratios above 0.3. Each run consisted of 1,000,000 ionic steps with a time step of 0.7 fs. The lowest-energy configurations obtained from these searches were subsequently fully relaxed using standard DFT calculations.

### AFM simulations
Simulated AFM images based on the DFT-relaxed structure (Fig. 1c,e) were generated using the Probe-Particle Model[70]. This approach accounts for the electrostatic potential above the surface (obtained from DFT calculations), Lennard-Jones potentials, and the elastic response of the tip. The lateral and vertical spring constants of the Cu tips ($k_{x,y} = 7.8\,N\,m^{-1}$, $k_z = 50.7\,N\,m^{-1}$) and the CuOx tips ($k_{x,y} = 161.9\,N\,m^{-1}$, $k_z = 271.1\,N\,m^{-1}$) were adopted from ref. 42. The effective tip charges were set to +0.05 e (Cu tip) and −0.05 e (CuOx tip), consistent with refs. 42,46–48. The simulated contrast was qualitatively unchanged with variations of the tip charge magnitude. The oscillation amplitude in each simulation matched the amplitude used in the corresponding experimental image. Because the exact tip–surface distance is not known experimentally, simulations were performed for a range of tip heights. The simulated images shown correspond to the tip–sample separations that yielded the best agreement with the experimental contrast.

### Reporting summary
Further information on research design is available in the Nature Portfolio Reporting Summary linked to this article.

## Data availability
The data that support the findings of this study are available from the corresponding author upon request. DFT-optimized structure models are provided as Supplementary Data 1–6. Unprocessed experimental raw data are provided as Supplementary Data 7.

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

## Acknowledgements
The computational results have been achieved using the Austrian Scientific Computing (ASC) infrastructure.

## Author contributions
J.B. conceived the research project. J.I.H-R. acquired and analyzed the experimental data. A.C. performed the computational modeling. J.I.H-R. and D.K. developed the sample preparation protocol. M.S. and F.M. supported computational modeling and data analysis. J.B. and M.S. supported experimental data acquisition and analysis. U.D. acquired funding. J.B. and U.D. supervised the project. J.I.H-R., A.C., U.D. and J.B. wrote the manuscript with contributions from all authors.

## Funding
This work was supported by the European Research Council (ERC) under the European Union's Horizon 2020 research and innovation program (Grant Agreement No. 883395) and funded in part by the Austrian Science Fund (FWF) [10.55776/F8100]. The authors acknowledge TU Wien Bibliothek for financial support through its Open Access Funding Program.

## Competing interests
The authors declare no competing interests.
