## [Transparent Peer Review file · Nature Communications]

AFM imaging reveals the unreconstructed α -Al₂O₃(0001) surface to be inhomogeneous and rough

Corresponding Author: Dr Jan Balajka

Version 0:

Reviewer comments:

Reviewer #1

(Remarks to the Author)

The article studies the surface of the technologically highly relevant material alumina (Al₂O₃) with atomic precision. The authors claim that this is perhaps the first AFM study achieving atomic resolution on this surface. If true, this alone makes the work significant. The authors further claim to find surprising results that contradict the previous assumption of a flat, uniformly Al-terminated surface. Instead, they report a highly irregular surface morphology with only small patches of Al termination.

The experimental results are supplemented by DFT calculations showing relaxation of surface Al atoms downward in order to compensate for their undercoordination, which is not in itself surprising. The AFM contrast is rationalized using simulations based on the probe-particle AFM (PP-AFM) model with Lennard-Jones interactions and $\pm 0.05e$ point charges to represent Cu and CuO tips.

I should note that I am not an expert in Al₂O₃ or oxide surfaces in general, and therefore I cannot fully judge the importance of this work for the field. In particular, I cannot fully assess whether the authors' claim that this work fundamentally revises the prevailing understanding of the Al₂O₃ surface structure is justified.

Based on my limited background, I would expect undercoordinated Al atoms to be energetically unfavorable, and therefore I am not surprised by the finding that such configurations are unstable. In that sense, I find it somewhat surprising that the prevailing assumption in the literature was the existence of a stable, uniformly Al-terminated surface.

The authors discuss oxygen passivation and hydroxylation extensively, and mention discrepancies between theoretical predictions (which favor dissociative adsorption of water and hydroxyl termination) and experimental observations (which often show only limited hydroxylation). Personally, I find this discrepancy more surprising than the instability of Al-terminated surfaces. However, the connection between this discussion and the AFM/DFT results presented in this paper is not always entirely clear to me.

In the Methods section, the surface preparation involves boiling in water followed by annealing at 820-900 °C in UHV or low-pressure O₂. It might help the reader if this process were more clearly contextualized in the discussion. Boiling presumably leads to hydroxylation, while annealing causes dehydration, but it is not clear what the relevant kinetic barriers are or which surface terminations should realistically be expected after this preparation sequence.

Do the authors attempt to directly address hydrogen or hydroxyl passivation in their study? If so, this should be stated more clearly, along with the specific hypotheses being tested. Otherwise, the discussion of hydroxylation risks being confusing. I assume the discussion of hydroxylation here is only interpretative and would benefit from clearer separation between experimental evidence and speculative reconciliation with prior literature.

Overall, I feel that the results raise more questions than they answer. The experimental data shown in Fig. 2 are also somewhat difficult to interpret without additional guidance. For example, in Fig. 2g (the reconstructed $\sqrt{31} \times \sqrt{31}$ surface), the periodicity is not immediately obvious; adding unit cell outlines could help. Likewise, for an untrained reader it is difficult to understand what exactly Fig. 2a represents structurally (in terms of atomic structure). The reported corrugation (~ 6 Å) seems large - does it correspond to atomic layers, or something else? Are these terraces, or something more disordered? Is the surface fully amorphous (hard to believe), or does it have some periodicity or structure on a large scale?

It is also not immediately clear how Fig. 2a and Fig. 2b relate to each other (e.g., is Fig. 2b a zoom-in of a region in Fig. 2a?). Otherwise it would help to see the same zoom area (Fig. 2b) in topographic view to see clearly corrugation in terms of atomic layers. Including such clarification would significantly improve readability.

The authors do not appear to provide a clear atomistic interpretation of this corrugation, and the DFT results in Fig. 1 do not directly address it. The MLFF simulations discussed in the Methods could potentially provide insight here, but these results are only shown in Extended Data Fig. 8, which is easy to miss. I would encourage the authors to bring these results more prominently into the main text.

Finally, regarding the PP-AFM simulations shown in Fig. 2e,f: these appear to be performed at relatively large tip-sample distances, likely in a regime dominated by electrostatic interactions rather than Pauli repulsion. In that case, the contrast may largely reflect the electrostatic potential (or its vertical derivative). It would be useful to clarify this explicitly. Showing maps of the electrostatic potential or E_z component (at least in supplementary) could help support this interpretation.

Authors mention that simulated height most similar to experiment was chosen. Perhaps the whole simulation range can be shown in supplementary, if experimental references (different heights) are available (especially data at closer tip-sample separation where Pauli repulsion starts to play a role). This would significantly help to clearly identify atoms and their corrugation. But I assume such experimental data are not available.

Additionally, the use of fixed point charges ($\pm 0.05e$) for the Cu and CuO tips is somewhat ad hoc. More advanced approaches, such as FDBM (available in PPAFM package as well) using ab initio charge densities for the tip, might offer a more realistic description (although specific shape of Cu cluster may bring unnecessary complication). That said, if the goal of the simulation is merely to demonstrate that the contrast arises from electrostatics and identifies Al sites as positively charged, then the current approach is sufficient.

Overall, I find this to be a valuable and technically strong paper that combines multiple complementary techniques to shed light on the surface structure of technologically important Al_2O_3 . However, due to the complexity of the system and the limited space for discussion, some aspects of the interpretation remain difficult to follow, and several important connections are only briefly or implicitly addressed.

Reviewer #2

(Remarks to the Author)

The manuscript by Huetner-Reisch et al. reports a structural analysis of the unreconstructed $\alpha\text{-Al}_2\text{O}_3(0001)$ surface. Previously the same group investigated the corresponding reconstructed surface, which was published in their Science paper from 2024. In both studies nc-AFM, XPS, LEED and advanced theoretical modeling are applied. In the present study, a highly corrugated surface is found, which usually is very challenging in terms of elucidating the surface structure with atomic resolution. Nevertheless, the authors were able to perform AFM experiments even with functionalized tips allowing to provide new insights to the local atomic structure of the surface, which is relevant for several research fields. The paper is generally very well written, however in my eyes, some of the conclusions are not fully supported by the presented data. Furthermore, I have several minor comments as listed below. After revision, the paper has the potential for a publication in Nature Communications.

Line 85-92: The way of argument how the authors establish “chemical identity of these locally Al-terminated regions ...” by imaging with two different tip terminations is not fully convincing. The Tip charge in the simulations appears randomly set. Setting the one tip positive and the other negative in the simulations basically determines the outcome. It is not clear where the information about tip charge for these tips is based on (at least I could not find it in ref 34). Therefore, the argument that the contrast is based on electrostatic interaction is not valid (line 88) and the conclusion that the dark depressions are Al is only weakly supported. Comparing experimental with DFT-simulated force-distance curves would be more convincing as opposed to the PPM model.

In Figure 2: the authors generally show no height dependent AFM data. Only single (optimized?) contrasts are shown. As a consequence it is not possible to assess in what height regime the data are recorded. Height dependent images or force spectroscopy data from the different sites would certainly strengthen the conclusions. Furthermore, this would allow excluding any dominating height dependent changes in the observed contrast.

In the introduction, the relevance of the presented study is motivated by citing numerous articles, most of which are published more than 15 years ago. To demonstrate the topicality of this work, discussing more recent works would help (if possible).

Line 32-33: “Throughout this transformation “ Statement needs a citation or justification

Fig.1.: nearly all presented data in Fig. 1 are taken from the previous Science paper published by the group. Although a reference is given at the end of Figure caption d, the authors should include appropriate permissions. Of course this holds also for the case of all other figures where applicable

Ext Data Fig. 1: Please add a line profile of the data in d to quantitatively assess the roughness easier.

Ext Data Fig. 4/5: “... the spectra were normalized to equal O 1s peak areas ...” not clear how this normalization has been done. For example, the peak areas in b are clearly different. Or is it a background normalization? Please clarify.

Ext Data Fig. 5: “The nearly unchanged Al:O ratio indicates that the unreconstructed and reconstructed surfaces have similar stoichiometry.” It is strongly suggested to quantitatively determine the $[\text{Al}]/[\text{O}]$ concentration ratios for both cases by applying the proper sensitivity factors to the peak area.

Line 72 It is recommended to include a brief sample preparation procedure here in the main text together with a statement that this is the procedure also used in literature to obtain the unreconstructed surface.

Line 76: here I think it would be important to make a statement that the authors believe (if so) that this allows to exclude any implications on the interpretation.

Fig 1a, please include a line profile allowing to quantify roughness better

Fig. 2a,b please indicate what kind of tip has been used for these data.

Ext Data Fig 6: please consider to include this Figure in the main manuscript

Extended Data Fig 7: data are only shown for the reconstructed surface and not for the unreconstructed case, which would be more relevant for the present paper.

Lines 148-153: speculating about the origin of the observed roughness, the authors employ DFT simulations of atomic steps as one possibility. It remains unclear how this would relate to the apparently highly defined step edges as found by large scale AFM imaging (Extend Data Fig 1 as compared to Fig 2a).

How about local charging effects as origin for the observed inhomogeneity and missing order? Can such an effect be excluded? In my eyes discussing this possibility would be appropriate.

Reviewer #3

(Remarks to the Author)

The manuscript "The Unreconstructed α -Al₂O₃(0001) Surface is Inhomogeneous and Rough" by Johanna Hütner-Reisch et al. describes a profound and challenging experimental study on the atomic structure of one of the most important crystal surfaces for fundamental understanding of surface physics and for various technological applications. The results are based on sound experimental procedures, are highly relevant and are generally very well supported by extended data and by calculations. In addition, the narrative is appealing, well written, and adequately supported by literature.

I can therefore heartily recommend publication of the manuscript in Nature Communications after clarification of the few remarks below and according minor revision of the manuscript.

1) On line 137-139, the authors compare the quality of obtained LEED patterns to previous work. I agree with the reasoning that LEED penetrates a few atomic layers and therefore represents also the bulk (1x1) structure. This is also observed, e.g., in the LEED patterns of the $(\sqrt{31} \times \sqrt{31})R\pm 9^\circ$ reconstructed surface. Still, I would expect superposition with a significant diffuse background coming from the disordered regions between the ordered Al-terminated islands. These disordered regions cover more than half of the surface (Fig. 2). Furthermore, I'd expect this background should be even more strongly observed in Reflection High-Energy Electron Diffraction (RHEED) studies, which is even more surface sensitive. This is not typically the case in the literature. Can the authors comment on the absence of such a diffuse background?

2) I am puzzled by the mechanism for forming such a rough surface. If the step edges have low formation energy, they should also be quite mobile and thereby, readily form larger domains at elevated temperatures; either by lateral movement or desorption from the step edges to eliminate the islands altogether. The observations contradict this scenario: the surface morphology is rather stable until the $(\sqrt{31} \times \sqrt{31})R\pm 9^\circ$ starts to form. This suggests the existence of a supercell of a stepped surface with lower surface energy density than the continuous (1x1) relaxed structure. Can the authors comment on such a possibility?

3) In the introduction (line 36-37), the authors refer to the (1x1) surface being the standard model for "epitaxial growth, adsorption, and surface chemistry studies." In the discussion (line 140 onward), they mostly refer to implications (water) adsorption and surface chemistry. Can the authors comment (briefly) on implications for epitaxial growth as well? I reckon that the implications of the reported inhomogeneity and roughness for epitaxial growth are severe, especially in the early stages of the growth. For example, layer-by-layer growth should be nearly impossible on such surfaces without the use of smoothening buffer layers.

4) Extended Data Figs. 4 and 5 show that the peak area of Al 2p is low compared to the O 1s peak at the surface of both the (1x1) and $(\sqrt{31} \times \sqrt{31})R\pm 9^\circ$ reconstruction. In Fig. 4, "The pronounced decrease in Al 2p intensity at 70° indicates a lower surface concentration of aluminum relative to the bulk." whereas in the caption of Fig. 5, "The nearly unchanged Al:O ratio indicates that the unreconstructed and reconstructed surfaces have similar stoichiometry". These two statements suggest that both surfaces contain less Al than the bulk. Considering the typical XPS penetration depth of a few nm (~1 nm for 70° incidence), more than just the very top layer is probed. How can this observation be understood from the perspective of a fully stoichiometric surface reconstruction in both the (1x1) and $(\sqrt{31} \times \sqrt{31})R\pm 9^\circ$ reconstructions?

- Very minor remarks concerning presentation

5) The authors explicitly mention the different imaging conditions very well in most of the work, except the tip that was used for recording the image in Fig. 2b. The overlay of the (1x1) unit cell suggests the bright spots to represent Al atoms as in Fig. 2c and thus I assume a Cu tip to have been used. Could the authors confirm and add this information to the manuscript?

6) Figs. 2(a) and 2(g) appear to have the same field-of-view but a different scale bar. It would help direct comparison to use either the 5-nm bar or the 2-nm in both. Perhaps, they can also be placed directly next to each other (vertically) for even

easier comparison.

Reviewer #4

(Remarks to the Author)

The manuscript “the unreconstructed α -Al₂O₃(0001) surface is inhomogeneous and rough” discusses the stability of unreconstructed and reconstructed surface of α -Al₂O₃. The nc-AFM and DFT calculations were combined to illuminate the surface geometry. This work is extension research of the same group’s previous work published on science. The major conclusion is that the unreconstructed (1 × 1) lattice is only a metastable state, which would transfer to a reconstructed surface under high temperature. However, this statement has already been proposed by their previous work and several other articles. The authors failed to give a clear explanation for the generation of nanometer-scale (1 × 1) lattice. According to the novelty and completeness, this work does not fulfill the standard of publishing on Nature Communications. I recommend to transfer the article to other journals such as Scientific Reports.

Comments on the manuscript

1. The authors argued that Al cations appeared as bright (repulsive) with a Cu tip. The same sites appeared as dark (attractive) with a CuOx tip. The brightness is not the only clue for the direction of interaction force. The $\Delta f \sim Z$ curve can be implemented to find the nature of the interaction.
2. The mechanism of generating metastable (1 × 1) lattice is not well studied in the manuscript. The concentration of oxygen does not affect the surface geometry since the reconstructed phase is thermodynamically stable with a large range of oxygen chemical potentials. Discussion is required for the appearance of the unreconstructed structure. Is this structure relative to defect, step or stress? In addition, the measured lattice parameters should be given for the images in Figure 2.
3. The authors think the unreconstructed surface is rough. This expression is not accurate since the so-called unreconstructed surface in Figure 2a is not a pure lattice. It contains multiple nanoscale lattices and other reconstructed structures.
4. The thermodynamic stability of the unreconstructed lattice should be studied. Simulating a larger supercell with small structural perturbations can verify the stability.
5. The proportion of elements is different between two types of surfaces. The calculation for the surface energy should be very careful especially when oxygen is involved. Some necessary detail is not shown in the computational methods part. Is the zero-point energy included in the calculation? Does the calculation setup gives correct energy difference between the singlet and triplet states of O₂ molecule.

Reviewer #5

(Remarks to the Author)

The authors present a combined noncontact atomic force microscopy and density functional theory study elucidating the atomic structure of the α -Al₂O₃(0001) surface. They reveal a rough and disordered morphology, with only nanometer-scale regions exhibiting the ordered Al-terminated (1×1) structure. These results provide valuable insights for the surface science community, particularly given the widespread use of α -Al₂O₃(0001) as a substrate for thin-film growth and as a support in heterogeneous catalysis.

While the study is of clear interest, several important issues should be addressed to strengthen its robustness and impact:

1. Thermodynamic treatment of coexisting surface domains.

In Fig. 2a, it appears that approximately 50% of the surface is covered by (1×1) domains, with the remainder being disordered regions. Although Fig. 1d indicates that the (1×1) surface is less stable than the reconstructed surface upon high-temperature annealing, the calculations appear to consider only a pure (1×1) phase. Given the experimental observation of coexisting ordered and disordered domains, their coexistence may alter the thermodynamic landscape and potentially stabilize the surface relative to a fully reconstructed phase. A more rigorous treatment would ideally consider mixed ordered/disordered surface configurations, even though modeling disorder is technically challenging.

2. Nature and origin of the disordered domains.

What is the atomic nature of the disordered regions? Do they contain both Al and O species, or are they dominated by one? Given their large surface coverage (~50%), there must be a thermodynamic driving force for their formation. A clearer microscopic picture of the mechanisms responsible for forming and stabilizing these disordered domains would significantly strengthen the interpretation of the experimental observations.

3. Annealing conditions in Extended Data Fig. 2.

What is the annealing environment in Extended Data Fig. 2: vacuum or O₂? If O₂ is used, what is the pressure? How do these conditions differ from the air annealing reported in Extended Data Fig. 1?

4. Annealing conditions in Extended Data Fig. 5.

Similarly, please clarify the annealing environment in Extended Data Fig. 5 (vacuum vs O₂), including the O₂ pressure if applicable, and how it compares with the air annealing conditions in Extended Data Fig. 1.

5. Role of O₂ in Extended Data Fig. 6.

In Extended Data Fig. 6, annealing is performed in 1×10⁻⁶ mbar O₂. Why was O₂ used instead of vacuum, as in Extended Data Fig. 2? Is O₂ required to drive the transition from (1×1) to the reconstructed surface, and if so, why? Is there any

difference in the transition temperature or pathway between O₂ and UHV annealing? If the surface does not lose lattice oxygen during annealing, the role of O₂ in this process should be clarified.

Addressing these points would significantly enhance the mechanistic understanding and general impact of the work.

Version 1:

Reviewer comments:

Reviewer #1

(Remarks to the Author)

I am satisfied with the authors' responses to my questions and comments, as well as with the revisions made to the manuscript. In particular, the improved presentation and layout of Fig. 2 significantly enhance the clarity and make the experimental results much easier to interpret.

In its current form, I consider the manuscript to be technically strong and suitable for publication.

That said, I do not feel sufficiently qualified to assess the broader significance of this work within the field of covalent oxide surfaces. In particular, other reviewers have raised concerns regarding the proximity of this study to the authors' previous publication in Science. I prefer not to take a position on this issue and would defer the evaluation of the work's impact and novelty within the field to reviewers with more specialized expertise in this area.

Reviewer #2

(Remarks to the Author)

The manuscript by Hütner-Reisch et al. was evaluated by 5 reviewers and the revisions made by the authors considerably improved it. Still, I found that my major concern that the argumentation how the authors establish chemical selectivity with two different tip terminations, is not sufficiently addressed. The authors argue that electrostatic interactions govern the contrast, but as said before, setting the one tip positive and the other negative in the simulations basically determines the outcome. Therefore, the choice of positive/negative tip charge has to be justified in more detail. Ref. 42. provides not sufficient support to assume a negative tip charge for CuO-tip and the positive charge at the metallic apex is also just assumed. Due to the importance of this issue for the main conclusions of the paper, a more substantial argumentation is needed. I recommend considering the following papers to do so: (1) 10.1088/1367-2630/ab8efd (2) 10.1021/acsnano.4c03155 (3) 10.1021/acs.nanolett.5b05251

Reviewer #3

(Remarks to the Author)

The authors have carefully addressed my comments and those of the other reviewers. The changes and additions to the manuscript have taken away my concerns and I can recommend publication of the manuscript in its current form.

Reviewer #4

(Remarks to the Author)

The manuscript is greatly improved after revision. Although I am not fully convinced that "the Al-terminated (1 x 1) surface cannot be experimentally obtained", this manuscript still reveals new fact to the terminating surface of Al₂O₃. I would recommend to publish this article on Nature communications.

If possible, the authors can give some effort on further improvement of this article regarding the following issues.

1. The authors use a reconstructed surface model with steps to argue that large-scale (1 x 1) structure is not stable. But the results show that steps are not energetically favored compared to the bare surface. The readers would expect some disordered structures (vacancies, adatoms, stacking faults, steps, et al) giving lower formation energy than the surface. In addition, the XPS spectra shows reduced Al-O ratio. This ratio should also be provided in their (1 x 1) ordered and disordered structures.
2. The authors switched the position of panel g and panel c-f in figure 2. I can not find the meaning of this treatment. On line 139, The sentence "Above ≈1000 °C, the surface transformed into an atomically flat and well-ordered ($\sqrt{31} \times \sqrt{31}$)R±9° reconstruction (Fig. 2b)." The authors changed Fig. 2g to Fig. 2b in the revised manuscript. This seems to be a typo.

Reviewer #5

(Remarks to the Author)

The authors have carefully addressed the comments and concerns raised in the previous review. The revisions have significantly improved the clarity, rigor, and overall quality of the manuscript. In particular, the additional analyses and expanded discussions provide a more balanced and convincing presentation of the results and their implications. I believe the manuscript is now suitable for publication in its current form and recommend acceptance.

Version 2:

Reviewer comments:

Reviewer #2

(Remarks to the Author)

My last comments have been reasonably addressed. I have no further concerns.

Reviewer #4

(Remarks to the Author)

All comments have been addressed properly. I recommend to publish this article on Nature Communicatiosn in current form.

Response to Reviewers of

The Unreconstructed α -Al₂O₃(0001) Surface is Inhomogeneous and Rough

Johanna I. Hütner-Reisch, Andrea Conti, David Kugler, Florian Mittendorfer, Michael Schmid, Ulrike Diebold, Jan Balajka

Review round: 1

We thank the Reviewers for their careful reading of our manuscript and for their constructive feedback. We have revised the manuscript following their suggestions, as detailed below.

Response to Reviewer 1

The article studies the surface of the technologically highly relevant material alumina (Al₂O₃) with atomic precision. The authors claim that this is perhaps the first AFM study achieving atomic resolution on this surface. If true, this alone makes the work significant. The authors further claim to find surprising results that contradict the previous assumption of a flat, uniformly Al-terminated surface. Instead, they report a highly irregular surface morphology with only small patches of Al termination.

The experimental results are supplemented by DFT calculations showing relaxation of surface Al atoms downward in order to compensate for their undercoordination, which is not in itself surprising. The AFM contrast is rationalized using simulations based on the probe-particle AFM (PP-AFM) model with Lennard-Jones interactions and $\pm 0.05e$ point charges to represent Cu and CuO tips.

I should note that I am not an expert in Al₂O₃ or oxide surfaces in general, and therefore I cannot fully judge the importance of this work for the field. In particular, I cannot fully assess whether the authors' claim that this work fundamentally revises the prevailing understanding of the Al₂O₃ surface structure is justified.

Based on my limited background, I would expect undercoordinated Al atoms to be energetically unfavorable, and therefore I am not surprised by the finding that such configurations are unstable. In that sense, I find it somewhat surprising that the prevailing assumption in the literature was the existence of a stable, uniformly Al-terminated surface.

Author response: We thank the Reviewer for recognizing the significance of our work. We agree that the widespread assumption of a stable, uniformly Al-terminated (1 × 1) surface is, in retrospect, surprising. This simple bulk termination, guided by the surface non-polarity considerations, and broadly adopted as the structural basis for interpreting the alumina (0001) surface, has not been challenged due to the lack of spatially resolved data capable of revealing the surface structure and its lateral heterogeneity. The images presented here build on advances in noncontact AFM that enabled atomically resolved imaging of insulating surfaces. In addition, our previous work on the reconstructed Al₂O₃(0001) surface (ref. 1) showed the relative instability of the (1 × 1) surface across the entire range of accessible oxygen chemical potentials. Taken together, we feel the need to revise the commonly assumed view of the unreconstructed alumina surface.

The authors discuss oxygen passivation and hydroxylation extensively, and mention discrepancies between theoretical predictions (which favor dissociative adsorption of water and hydroxyl termination) and

experimental observations (which often show only limited hydroxylation). Personally, I find this discrepancy more surprising than the instability of Al-terminated surfaces. However, the connection between this discussion and the AFM/DFT results presented in this paper is not always entirely clear to me.

Author response: The core of the discrepancy between theoretical predictions of dissociative water adsorption and experimental observations of limited hydroxylation is that the predictions are based on an idealized structure model that is not representative of the majority of the surface. The Al-terminated (1×1) surface is predicted to promote facile water dissociation. However, our images show that this termination occurs only in small patches, while most of the surface lacks these exposed reactive Al sites. To clarify this connection, the introduction paragraph discussing reactivity toward water in the revised manuscript now contains an additional sentence: “*Here, we show that the disagreement between theory and experiment arises from the fact that the majority of the unreconstructed $Al_2O_3(0001)$ surface differs from the commonly assumed idealized structure model.*”

In the Methods section, the surface preparation involves boiling in water followed by annealing at 820-900 °C in UHV or low-pressure O_2 . It might help the reader if this process were more clearly contextualized in the discussion. Boiling presumably leads to hydroxylation, while annealing causes dehydration, but it is not clear what the relevant kinetic barriers are or which surface terminations should realistically be expected after this preparation sequence.

Do the authors attempt to directly address hydrogen or hydroxyl passivation in their study? If so, this should be stated more clearly, along with the specific hypotheses being tested. Otherwise, the discussion of hydroxylation risks being confusing. I assume the discussion of hydroxylation here is only interpretative and would benefit from clearer separation between experimental evidence and speculative reconciliation with prior literature.

Author response: Our study does not directly address hydrogen or hydroxyl passivation. The discussion of hydroxylation is included to reconcile prior literature in the context of our findings. The boiling step was performed as a cleaning procedure to remove soluble contaminants from the crystals and sample plates before introducing them to UHV. In our experience, without boiling in ultrapure water, high-temperature annealing can promote diffusion of trace impurities (*e.g.*, alkali metals), which may lead to mixed surface phases and potential misinterpretation of results. The boiling step was therefore included to minimize contamination, not to deliberately induce hydroxylation. The results discussed in the manuscript (Fig. 2 and Supplementary Figs. 2,3,5,7–9) were obtained after annealing above 800 °C in UHV or low O_2 partial pressure and correspond to a non-hydroxylated surface, as evidenced by the absence of an OH-related signal in XPS (Supplementary Fig. 7). We have revised the Methods section to clarify the sample preparation procedure, and included a comment on the absence of hydroxylation detected by XPS in the figure caption of Supplementary Fig. 7 to better separate our results from reconciliation of prior hydroxylation studies.

Overall, I feel that the results raise more questions than they answer. The experimental data shown in Fig. 2 are also somewhat difficult to interpret without additional guidance. For example, in Fig. 2g (the reconstructed $\sqrt{31} \times \sqrt{31}$ surface), the periodicity is not immediately obvious; adding unit cell outlines could help. Likewise, for an untrained reader it is difficult to understand what exactly Fig. 2a represents structurally (in terms of atomic structure). The reported corrugation ($\sim 6 \text{ \AA}$) seems large - does it correspond to atomic layers, or something else? Are these terraces, or something more disordered? Is the surface fully amorphous (hard to believe), or does it have some periodicity or structure on a large scale?

Author response: We thank the Reviewer for the suggestions regarding the visual readability of Fig. 2 and agree that the inherent structural complexity challenges the simplified view of this surface. We have modified Fig. 2 to improve clarity. The topography image of the ($\sqrt{31} \times \sqrt{31}$) reconstructed surface in Fig. 2g and the atomically resolved inset are no longer rotated relative to each other; the outlined unit cells (red rhombi) therefore now have the same orientation. The topography image in Fig. 2a shows the morphology of the unreconstructed surface within a single macroscopic terrace. To aid interpretation, we have added a representative line profile. The reported vertical range of $\sim 6 \text{ \AA}$ corresponds to approximately three monoatomic step heights (216 pm each). However, no well-defined atomic step edges or extended periodic features were observed on this highly irregular and rough surface. The images indicate a laterally inhomogeneous surface with only local atomic order (as resolved in Fig. 2b), rather than a fully amorphous structure or a well-ordered terrace-step morphology. Below, we include a Fast Fourier transform (FFT) of the image in Fig. 2a to demonstrate the absence of periodic features in the image. The Results section in the revised manuscript has been modified accordingly to clarify this interpretation.

Fig. R1. The unreconstructed $\text{Al}_2\text{O}_3(0001)$ surface lacks long-range order. (a) Topography image corresponding to Fig. 2a in the manuscript. (b) Fast Fourier transform (FFT) of the image in (a). The FFT does not exhibit sharp, well-defined features indicative of extended periodic order. The vertically aligned spots are due to instrumental noise of the nc-AFM.

It is also not immediately clear how Fig. 2a and Fig. 2b relate to each other (e.g., is Fig. 2b a zoom-in of a region in Fig. 2a?). Otherwise it would help to see the same zoom area (Fig. 2b) in topographic view to see clearly corrugation in terms of atomic layers. Including such clarification would significantly improve readability.

Author response: Fig. 2b is a zoom-in of a region in Fig. 2a, acquired in constant-height mode to resolve the local atomic order. Fig. 2a was recorded in constant-frequency-shift mode to capture the larger-scale surface morphology. To clarify the relationship between the two panels, we have indicated the position and frame size of the high-resolution image (Fig. 2b) within the corresponding topography image (Fig. 2a).

The authors do not appear to provide a clear atomistic interpretation of this corrugation, and the DFT results in Fig. 1 do not directly address it. The MLFF simulations discussed in the Methods could potentially provide insight here, but these results are only shown in Supplementary Fig. 8, which is easy to miss. I would encourage the authors to bring these results more prominently into the main text.

Author response: We thank the Reviewer for this suggestion. The computational modeling of steps is intended to provide a rationale for the observed rough morphology, based on an estimate of the step energy, rather than to propose an atomistic model of the unreconstructed surface. The calculations show that forming stepped configurations is energetically inexpensive, because such configurations allow locally higher coordination of surface Al atoms (by forming O-terminated motifs) compared to a flat, Al-terminated (1×1) structure. The structure shown in Supplementary Fig. 8 (now Supplementary Fig. 12) is therefore presented as an illustrative example of a low-energy stepped configuration, not as a structural solution of the experimentally observed surface. Given that constant-height nc-AFM primarily probes the most protruding atoms, a complete atomistic description of the disordered regions is not experimentally accessible in the present work. We have expanded the Discussion section and the figure caption of Supplementary Fig. 12 (previously 8) in the revised the manuscript to more clearly emphasize this point.

Finally, regarding the PP-AFM simulations shown in Fig. 2e,f: these appear to be performed at relatively large tip-sample distances, likely in a regime dominated by electrostatic interactions rather than Pauli repulsion. In that case, the contrast may largely reflect the electrostatic potential (or its vertical derivative). It would be useful to clarify this explicitly. Showing maps of the electrostatic potential or E_z component (at least in supplementary) could help support this interpretation.

Authors mention that simulated height most similar to experiment was chosen. Perhaps the whole simulation range can be shown in supplementary, if experimental references (different heights) are available (especially data at closer tip-sample separation where Pauli repulsion starts to play a role). This would significantly help to clearly identify atoms and their corrugation. But I assume such experimental data are not available.

Author response: We agree with the Reviewer's interpretation. The PP-AFM simulations were performed at relatively large tip-sample distances, in a regime dominated by electrostatic interactions. Experimentally, approaching the tip closer resulted in increased interaction with the undercoordinated surface Al atoms and imaging instabilities before entering the Pauli repulsion regime. The simulations were therefore carried out at distances corresponding to stable experimental imaging conditions. To clarify this point, we now include maps of the electrostatic potential and the E_z component in Supplementary Fig. 6. In addition, we now provide height-dependent experimental images together with corresponding PP-AFM simulations over the same range of tip-sample separations in Supplementary Fig. 5. In both experiment and simulation, the contrast remains qualitatively unchanged over this height range, consistently identifying the surface Al atoms as attractive to the O-terminated CuOx tip. This height invariance supports the conclusion that the contrast originates predominantly from electrostatic interactions.

Additionally, the use of fixed point charges ($\pm 0.05e$) for the Cu and CuO tips is somewhat ad hoc. More advanced approaches, such as FDBM (available in PPAFM package as well) using ab initio charge densities for the tip, might offer a more realistic description (although specific shape of Cu cluster may bring unnecessary complication). That said, if the goal of the simulation is merely to demonstrate that the contrast arises from electrostatics and identifies Al sites as positively charged, then the current approach is sufficient.

Author response: The effective point charge of $-0.05e$ for the CuOx tip was adopted from the literature (doi: 10.1039/D1NR04080D). For the Cu-terminated tip, no published value is available; we therefore selected a small positive charge of comparable magnitude, consistent with commonly used parameters in PP-AFM simulations. We verified that varying the charge magnitude does not qualitatively affect the simulated contrast and the identification of the surface Al atoms as the dominant electrostatic interaction sites. As the Reviewer correctly notes, the purpose of the simulations is to demonstrate that the experimentally observed contrast arises from electrostatics and to identify the surface atoms as positively charged Al cations. We now include an nc-AFM image of the Cu(110)-(2 × 1)-O surface acquired with the same tip in Supplementary Fig. 4 in the revised manuscript. This “fingerprint” image serves as a verification for the CuOx tip termination and allows identification of the attractive species as Al cations. We also explored the Full Density-Based model (FDBM, doi: 10.1016/j.cpc.2024.109341) approach using ab-initio charge densities. However, this method requires a structure model of the tip, which is not known. In addition, periodic boundary conditions can introduce artificial interactions due to long range electric fields associated with asymmetric tip models. Given these additional uncertainties, and since the simpler point-charge model reproduces the experimental contrast and its height dependence, we consider it appropriate for interpreting the image contrast in this study.

Overall, I find this to be a valuable and technically strong paper that combines multiple complementary techniques to shed light on the surface structure of technologically important Al₂O₃. However, due to the complexity of the system and the limited space for discussion, some aspects of the interpretation remain difficult to follow, and several important connections are only briefly or implicitly addressed.

Author response: We thank the Reviewer for recognizing the technical strength of our work. We appreciate the constructive suggestions aimed at improving the clarity of the manuscript and have revised the text accordingly to better connect the different observations and make the interpretation easier to follow.

Response to Reviewer 2

The manuscript by Huetner-Reisch et al. reports a structural analysis of the unreconstructed alpha-Al₂O₃(0001) surface. Previously the same group investigated the corresponding reconstructed surface, which was published in their Science paper from 2024. In both studies nc-AFM, XPS, LEED and advanced theoretical modeling are applied. In the present study, a highly corrugated surface is found, which usually is very challenging in terms of elucidating the surface structure with atomic resolution. Nevertheless, the authors were able to perform AFM experiments even with functionalized tips allowing to provide new insights to the local atomic structure of the surface, which is relevant for several research fields. The paper is generally very well written, however in my eyes, some of the conclusions are not fully supported by the presented data. Furthermore, I have several minor comments as listed below. After revision, the paper has the potential for a publication in Nature Communications.

Author response: We thank the Reviewer for the positive assessment of our work and for recognizing its relevance. We appreciate the constructive comments and the opportunity to strengthen the manuscript in response to the points raised below.

Line 85-92 The way of argument how the authors establish “chemical identity of these locally Al-terminated regions” by imaging with two different tip terminations is not fully convincing. The Tip charge in the simulations appears randomly set. Setting the one tip positive and the other negative in the simulations basically determines the outcome. It is not clear where the information about tip charge for these tips is based on (at least I could not find it in ref 34). Therefore, the argument that the contrast is based on electrostatic interaction is not valid (line 88) and the conclusion that the dark depressions are Al is only weakly supported. Comparing experimental with DFT-simulated force-distance curves would be more convincing as opposed to the PPM model.

In Figure 2: the authors generally show no height dependent AFM data. Only single (optimized?) contrasts are shown. As a consequence it is not possible to assess in what height regime the data are recorded. Height dependent images or force spectroscopy data from the different sites would certainly strengthen the conclusions. Furthermore, this would allow excluding any dominating height dependent changes in the observed contrast.

Author response: The experimental identification of the surface Al atoms is based on a controlled CuOx tip functionalization protocol developed by Mönig *et al.* (see ref. 42 or doi: 10.1021/acsnano.4c03155). The CuOx termination was verified experimentally by a “fingerprint” image acquired with the same tip as Fig. 2e, on a partially O-covered Cu(110) reference sample. Such a reference image is now included as Supplementary Fig. 4. In the AFM simulations, the effective charge of $-0.05e$ for the CuOx tip was adopted from the published value (Schulze Lammers *et al.*, ref. 42). For the Cu-terminated tip, no published value is available; we therefore selected a small positive charge of comparable magnitude, consistent with commonly used parameters in probe-particle AFM simulations. However, the simulated image contrast is qualitatively unchanged within reasonable variations of the tip charge, and the identification of the surface sites as Al atoms does not depend on the exact charge value. The observed contrast inversion between negatively and positively terminated tips is consistent with an electrostatic interaction mechanism. To support this interpretation, we now include height-dependent experimental images together with the corresponding simulations (Supplementary Fig. 5). The contrast remains qualitatively unchanged over the accessible range of tip-sample distances. At smaller separations, tip-surface interactions lead to imaging

instabilities before entering the Pauli repulsion regime. The images shown in the manuscript were therefore acquired under stable imaging conditions dominated by electrostatic interactions. This interpretation is further supported by the DFT-calculated maps of electrostatic potential and vertical electric field above the surface (Supplementary Fig. 6), which reproduce the experimentally observed contrast. We have revised the Results section of the manuscript to clarify the interpretation of the image contrast.

In the introduction, the relevance of the presented study is motivated by citing numerous articles, most of which are published more than 15 years ago . To demonstrate the topicality of this work, discussing more recent works would help (if possible).

Author response: We thank the Reviewer for the suggestion. While the atomic structure of the unreconstructed $\text{Al}_2\text{O}_3(0001)$ has not been revisited in detail in recent years, the broad temporal span of the cited literature reflects that this question has remained a long-standing issue in surface science. Moreover, recent studies typically assume a flat and uniformly (1×1) -terminated surface as a model substrate for epitaxial growth and surface chemistry. To emphasize the present relevance of our work, we have included several recent references in the introduction that employ the unreconstructed $\text{Al}_2\text{O}_3(0001)$ surface for epitaxial growth (doi: 10.1038/s41565-023-01445-9, 10.1038/s41586-022-04523-5, 10.1038/s41565-023-01456-6, 10.1016/j.apmt.2021.100975) and for studies of water interaction (10.1021/acs.jpcc.2c03743, 10.1021/acs.jpcc.5b10695, 10.1016/j.cis.2017.12.004).

Line 32-33: “Throughout this transformation “ Statement needs a citation or justification

Author response: We thank the Reviewer for pointing out this potentially confusing statement. To avoid ambiguity, we have rephrased the sentence in the Introduction. The revised text now reads: “*The reconstruction preserves the overall stoichiometry, i.e., the composition of both the unreconstructed and the reconstructed surfaces remains Al_2O_3 .*”

Fig.1.: nearly all presented data in Fig. 1 are taken from the previous Science paper published by the group. Although a reference is given at the end of Figure caption d, the authors should include appropriate permissions. Of course this holds also for the case of all other figures where applicable

Author response: We thank the Reviewer for raising this point. All figures in the present manuscript were newly generated for this work and do not reproduce previously published figures, with the exception of the structure models shown in panels a and c of Fig. 1. These panels use the same visualization of the bulk Al_2O_3 lattice and the creation of a nonpolar (0001) surface that were previously published in the Supporting Materials of ref. 1. We have obtained the appropriate permissions from the publisher for reuse of these panels and have added the corresponding credit line to the figure caption. No other previously published figures or figure panels have been reused, and no additional permissions are required.

Supplementary Fig. 1: Please add a line profile of the data in d to quantitatively assess the roughness easier.

Author response: We thank the Reviewer for this suggestion. We have added a representative line profile to Supplementary Fig. 1d to allow a more quantitative assessment of the surface roughness.

Supplementary Fig. 4/5: “... the spectra were normalized to equal O 1s peak areas ...” not clear how this normalization has been done. For example, the peak areas in b are clearly different. Or is it a background normalization? Please clarify.

Author response: The O 1s spectra were normalized by dividing each spectrum by its integrated peak area (after background subtraction). The corresponding Al 2p spectrum measured under the same conditions was divided by the same normalization factor. As a result, the O 1s spectra shown in panels (a) of Supplementary Figs. 4 and 5 (now Supplementary Figs. 7 and 8) have identical peak areas (normalized to unity), whereas the Al 2p peak areas in panels (b) differ and reflect changes in stoichiometry or vertical distribution. For example, the reduced Al 2p intensity at the more surface-sensitive 70° emission angle (previous Supplementary Fig. 4b, now 7b) indicates a lower relative Al contribution near the surface, consistent with a predominantly oxygen-terminated surface. We have revised the captions of Supplementary Figs. 4 and 5 (now Supplementary Figs. 7 and 8) and included a detailed description of the XPS normalization procedure in the Methods section.

Supplementary Fig. 5: “The nearly unchanged Al:O ratio indicates that the unreconstructed and reconstructed surfaces have similar stoichiometry.” It is strongly suggested to quantitatively determine the [Al]/[O] concentration ratios for both cases by applying the proper sensitivity factors to the peak area.

Author response: We thank the Reviewer for this constructive suggestion. We have performed a quantitative evaluation of the peak areas using appropriate sensitivity factors, assuming that the analyzer detection efficiency is the same for the Al 2p and O 1s electron energies. The resulting [O]/[Al] ratios are as follows:

- Unreconstructed surface: 1.52 (0° emission), 2.41 (70° emission)
- Reconstructed surface: 1.59 (0° emission), 2.19 (70° emission)

The ratios obtained in normal emission (0°), which probe several atomic layers, are close to the stoichiometric value of 1.5 expected for Al₂O₃. At the more surface-sensitive 70° emission angle, the relative oxygen signal increases, consistent with an oxygen-terminated outermost layer and increased surface sensitivity of this geometry. Both reconstructed and unreconstructed surfaces yield comparable [O]/[Al] ratios at each emission angle, indicating that their overall stoichiometry is similar. Details of the quantitative analysis have been added to the Methods section of the revised manuscript and to the figure captions of Supplementary Figs. 7 and 8.

Line 72 It is recommended to include a brief sample preparation procedure here in the main text together with a statement that this is the procedure also used in literature to obtain the unreconstructed surface.

Author response: We have added a brief description of the sample preparation procedure to the Results section in the main text and clarified that that the preparation protocol is consistent with procedures commonly used in the literature to obtain the unreconstructed Al₂O₃(0001) surface. Appropriate references have been included.

Line 76: here I think it would be important to make a statement that the authors believe (if so) that this allows to exclude any implications on the interpretation.

Author response: We agree with the Reviewer and have clarified this point explicitly in the manuscript. XPS confirms that high-temperature annealing removed surface contamination. A minute fluorine signal was detected only on the sample shown in Fig. 2; however, this signal was not reproduced on other samples exhibiting the same morphology. We are therefore confident that this trace fluorine contamination does not

influence the interpretation of the reported results. We have added a corresponding statement to the Results section of the revised manuscript.

Fig 1a, please include a line profile allowing to quantify roughness better

Author response: We have added a line profile to Fig. 2a for a more quantitative assessment of the surface roughness.

Fig. 2a,b please indicate what kind of tip has been used for these data.

Author response: The images in Fig. 2a,b were acquired with a Cu-terminated tip. This information has now been included in the figure caption.

Supplementary Fig. 6: please consider to include this Figure in the main manuscript

Author response: We thank the Reviewer for this suggestion. We carefully considered whether to move Supplementary Fig. 6 to the main manuscript and decided to retain it as supplementary figure to preserve the focus and clarity of the main narrative. The primary aim of this work is to elucidate the unreconstructed surface, while the $(\sqrt{31} \times \sqrt{31})R\pm 9^\circ$ reconstructed phase is discussed mainly for context. We did not perform a systematic study of the transformation between these terminations; the data in Supplementary Fig. 6 (now Supplementary Fig. 9) are included to demonstrate the absence of temperature-induced coarsening of the (1×1) islands. We therefore believe that presenting this information as supplementary figure supports the conclusions without diverting from the central focus of the manuscript.

Supplementary Fig 7: data are only shown for the reconstructed surface and not for the unreconstructed case, which would be more relevant for the present paper.

Author response: Supplementary Fig. 7 (now Supplementary Fig. 10) is included to demonstrate the irreversibility and environmental robustness of the $(\sqrt{31} \times \sqrt{31})R\pm 9^\circ$ reconstruction once formed. Historically, the transition between the unreconstructed and reconstructed $\text{Al}_2\text{O}_3(0001)$ surfaces was believed to be reversible and controlled by the oxygen chemical potential (*e.g.*, doi: 10.1021/j100706a014), because the reconstructed surface was considered reduced. Our results instead show that both surfaces are stoichiometric and that the reconstruction is induced by increased atomic mobility at elevated temperature. Once formed, the reconstructed phase is irreversible and independent of the O_2 background. While we did not perform identical experiments to those shown in Supplementary Fig. 7 on the unreconstructed surface, we show that annealing above 1000 °C in 10^{-6} mbar O_2 induces the reconstruction. Previous studies have further demonstrated that annealing at sufficiently high temperature leads to formation of the reconstructed phase even at much higher O_2 pressures (doi: 10.1002/adma.202312899). Under the conditions shown in Supplementary Fig. 7a (1300 °C, 10^{-4} mbar O_2), the surface reconstruction is thus expected to form as well. Exposure of the unreconstructed surface to liquid water at room temperature (as in Supplementary Fig. 7b) was not performed. Nevertheless, such experiment at room temperature is not expected to lead to a rearrangement of the subsurface layer. LEED may not capture possible surface modifications induced by water exposure. Considering the poorly ordered surface, it is likely that the (1×1) pattern predominantly stems from the bulk lattice and does not reflect local structural changes of the rough and irregular surface layer. The focus of the present work is the metastability of the unreconstructed surface. Supplementary Fig. 7 (now Supplementary Fig. 10) complements this discussion by illustrating that, once formed, the reconstructed phase is thermodynamically and environmentally robust and does not revert to the

unreconstructed termination under the explored conditions, consistent with the substantial energetic stabilization shown in the phase diagram in Fig. 1d.

Lines 148-153: speculating about the origin of the observed roughness, the authors employ DFT simulations of atomic steps as one possibility. It remains unclear how this would relate to the apparently highly defined step edges as found by large scale AFM imaging (Supplementary Fig. 1 as compared to Fig. 2a).

Author response: The large-scale AFM images (Supplementary Fig. 1) show well-defined step edges originating from the miscut of the single crystal. However, the terraces *between* these steps exhibit considerable roughness, as apparent in Supplementary Fig. 1d, consistent with the highly corrugated morphology observed at higher resolution in Fig. 2a. We have added a line profile in Supplementary Fig. 1e to underline the presence of roughness. The stepped structure considered in the DFT calculations is intended as a simplified model for the roughness on the few-nm scale (Fig. 2a), not the miscut with step–step distances of hundreds of nanometers. The coexistence of straight macroscopic step edges with nanoscale roughness on the terraces may originate from the surface termination outside UHV. Samples exposed to ambient conditions are expected to form a hydroxylated surface termination. Subsequent dehydroxylation toward Al₂O₃ requires substantial mass transport and can lead to a rough morphology. We have clarified this point in the Discussion section of the revised manuscript.

How about local charging effects as origin for the observed inhomogeneity and missing order? Can such an effect be excluded? In my eyes discussing this possibility would be appropriate.

Author response: We did not observe effects consistent with local charging on the Al₂O₃(0001) surface. In our experience, charged insulating surfaces, for example cleaved aluminosilicates (doi: 10.1038/s41467-023-35872-y) produce strong long-range electrostatic contributions to the tip-sample interaction, manifested as large frequency shifts in nc-AFM already at large tip-sample distances (typically hundreds of Hz unless compensated). In contrast, the frequency shifts due to electrostatic forces observed here were below 20 Hz, well within the range commonly observed for uncharged surfaces including metals (see, for example doi: 10.1126/science.adq4744, 10.1126/sciadv.aea2378, or Supplementary Fig. 4). The Al₂O₃(0001) samples were annealed prior to measurement, which may promote dissipation of residual charges. Based on the absence of pronounced long-range electrostatic contributions in the nc-AFM data, local charging can be excluded as the origin of the observed lateral inhomogeneity.

Response to Reviewer 3

The manuscript “The Unreconstructed α -Al₂O₃(0001) Surface is Inhomogeneous and Rough” by Johanna Hütner-Reisch et al. describes a profound and challenging experimental study on the atomic structure of one of the most important crystal surfaces for fundamental understanding of surface physics and for various technological applications. The results are based on sound experimental procedures, are highly relevant and are generally very well supported by extended data and by calculations. In addition, the narrative is appealing, well written, and adequately supported by literature.

I can therefore heartily recommend publication of the manuscript in Nature Communications after clarification of the few remarks below and according minor revision of the manuscript.

Author response: We thank the Reviewer for the encouraging assessment of our work and for the constructive suggestions, which have helped improve the manuscript as detailed below.

1) On line 137-139, the authors compare the quality of obtained LEED patterns to previous work. I agree with the reasoning that LEED penetrates a few atomic layers and therefore represents also the bulk (1x1) structure. This is also observed, e.g., in the LEED patterns of the $(\sqrt{31} \times \sqrt{31})R_{\pm 9^\circ}$ reconstructed surface. Still, I would expect superposition with a significant diffuse background coming from the disordered regions between the ordered Al-terminated islands. These disordered regions cover more than half of the surface (Fig. 2). Furthermore, I'd expect this background should be even more strongly observed in Reflection High-Energy Electron Diffraction (RHEED) studies, which is even more surface sensitive. This is not typically the case in the literature. Can the authors comment on the absence of such a diffuse background?

Author response: Diffuse background is indeed present in our LEED patterns. The image contrast has been optimized to highlight the diffraction spots, which reduces the visual prominence of the diffuse background. In direct comparison, the (1 × 1) LEED pattern of the unreconstructed surface exhibits noticeably stronger diffuse intensity than the $(\sqrt{31} \times \sqrt{31})R_{\pm 9^\circ}$ pattern of the reconstructed surface (see Supplementary Fig. 2). We now include Supplementary Fig. 11, to quantitatively compare diffraction spot intensities and diffuse background in LEED images of the unreconstructed and reconstructed surfaces. This trend is consistent with RHEED data reported in the literature, where the (1 × 1) patterns of the unreconstructed surface show enhanced diffuse background relative to the reconstructed phase (e.g., doi: 10.1016/j.apsusc.2020.148548; 10.1016/j.apsusc.2025.162929; 10.1002/adma.202312899; 10.1134/S1063782615070180). We have included this discussion in the Discussion section of the revised manuscript.

2) I am puzzled by the mechanism for forming such a rough surface. If the step edges have low formation energy, they should also be quite mobile and thereby, readily form larger domains at elevated temperatures; either by lateral movement or desorption from the step edges to eliminate the islands altogether. The observations contradict this scenario: the surface morphology is rather stable until the $(\sqrt{31} \times \sqrt{31})R_{\pm 9^\circ}$ starts to form. This suggests the existence of a supercell of a stepped surface with lower surface energy density than the continuous (1x1) relaxed structure. Can the authors comment on such a possibility?

Author response: We thank the Reviewer for this thoughtful comment. The step formation energy reflects the energetic cost of creating a step relative to a flat surface, whereas step mobility is governed by activation

barriers that depend on local bonding configurations and do not necessarily scale with the step formation energy. In fact, low-energy steps are often associated with more stable and more strongly coordinated atomic configurations, which can exhibit higher activation barriers for rearrangement. Conversely, higher-energy steps may contain a higher density of kink sites and more weakly bound atoms, facilitating atom exchange and leading to higher mobility. In the present system, our calculations indicate that the energetic cost of forming locally stepped configurations is low relative to the flat Al-terminated (1×1) surface, indicating that the driving force towards flattening a rough surface is weak. At the same time, the low step energy implies that the atoms at the step occupy low-energy, strongly bound configurations, which reduces step mobility. The observed stability of the rough morphology up to the temperature at which the $(\sqrt{3}1 \times \sqrt{3}1)R\pm 9^\circ$ reconstruction forms suggests that large-scale mass transport is kinetically limited under these conditions. Once sufficiently high temperatures are reached, atomic mobility increases substantially and the system transitions to the energetically favored reconstructed phase, which provides a substantial reduction in surface energy (Fig. 1d). We have included this comment in the figure caption of Supplementary Fig. 12 in the revised manuscript.

3) In the introduction (line 36-37), the authors refer to the (1x1) surface being the standard model for “epitaxial growth, adsorption, and surface chemistry studies.” In the discussion (line 140 onward), they mostly refer to implications (water) adsorption and surface chemistry. Can the authors comment (briefly) on implications for epitaxial growth as well? I reckon that the implications of the reported inhomogeneity and roughness for epitaxial growth are severe, especially in the early stages of the growth. For example, layer-by-layer growth should be nearly impossible on such surfaces without the use of smoothening buffer layers.

Author response: We thank the referee for this important suggestion. We agree that the reported inhomogeneity and roughness of the unreconstructed $\text{Al}_2\text{O}_3(0001)$ surface have direct implications for epitaxial growth, especially during the early stages of film formation. In particular, for two-dimensional materials and ultrathin films, where the interface structure plays a critical role, such inhomogeneity may significantly affect growth behavior and film quality. For example, a recent study has demonstrated substantially larger crystal size and nucleation density of two-dimensional tungsten disulfide grown on the reconstructed surface (doi: 10.1039/D5NR03765D). We have now included a brief discussion of these implications for epitaxial growth in Discussion section of the revised manuscript.

4) Supplementary Figs. 4 and 5 show that the peak area of Al 2p is low compared to the O 1s peak at the surface of both the (1x1) and $(\sqrt{3}1 \times \sqrt{3}1)R\pm 9^\circ$ reconstruction. In Fig. 4, “The pronounced decrease in Al 2p intensity at 70° indicates a lower surface concentration of aluminum relative to the bulk.” whereas in the caption of Fig. 5, “The nearly unchanged Al:O ratio indicates that the unreconstructed and reconstructed surfaces have similar stoichiometry”. These two statements suggest that both surfaces contain less Al than the bulk. Considering the typical XPS penetration depth of a few nm (~ 1 nm for 70° incidence), more than just the very top layer is probed. How can this observation be understood from the perspective of a fully stoichiometric surface reconstruction in both the (1x1) and $(\sqrt{3}1 \times \sqrt{3}1)R\pm 9^\circ$ reconstructions?

Author response: The Reviewer correctly notes that the XPS data indicate that both the unreconstructed and the $(\sqrt{3}1 \times \sqrt{3}1)R\pm 9^\circ$ reconstructed surfaces are predominantly oxygen-terminated at the outermost layer, as reflected by the enhanced [O]/[Al] ratio at the more surface-sensitive 70° emission. Although XPS probes several atomic layers, the detected signal is attenuated with depth and the outermost layer contributes

most to the measured intensity. Calculations using the SESSA program (doi: 10.1002/sia.2097) yield effective mean free paths of 0.98 nm for O 1s and 1.3 nm for Al 2p, corresponding to 1/e decay lengths of 0.34 nm and 0.44 nm assuming a simple $\cos(\theta)$ dependence. Therefore, measurements in normal emission probe multiple layers and yield [O]/[Al] ratios close to the stoichiometric value of 1.5 expected for Al₂O₃ for both the reconstructed and unreconstructed surfaces (see also our response to Reviewer 2). In contrast, measurements at 70° emission are significantly more surface sensitive. The increased [O]/[Al] ratio at this emission angle therefore reflects oxygen termination of the surface rather than a deviation from overall stoichiometry. Using the reconstructed surface as a reference with established stoichiometry, we conclude that both surfaces are overall stoichiometric, with a predominantly oxygen-terminated outermost layer. We note that photoelectron diffraction effects may additionally contribute to the reduced Al signal at 70° emission. However, the similar [O]/[Al] ratios observed for both surfaces at this emission angle indicate comparable average surface composition and support similar near-surface structural arrangements. We now provide quantitative analysis of the peak areas in the figure captions of Supplementary Figs. 7 and 8.

- Very minor remarks concerning presentation

5) The authors explicitly mention the different imaging conditions very well in most of the work, except the tip that was used for recording the image in Fig. 2b. The overlay of the (1x1) unit cell suggests the bright spots to represent Al atoms as in Fig. 2c and thus I assume a Cu tip to have been used. Could the authors confirm and add this information to the manuscript?

Author response: The images in Fig. 2b and 2c were acquired with a Cu-terminated tip. This information has been added to the figure caption.

6) Figs. 2(a) and 2(g) appear to have the same field-of-view but a different scale bar. It would help direct comparison to use either the 5-nm bar or the 2-nm in both. Perhaps, they can also be placed directly next to each other (vertically) for even easier comparison.

Author response: We have unified the scale bars in Figs. 2a and 2g to 2 nm and rearranged the figure so that the two images are placed directly next to each other.

Response to Reviewer 4

The manuscript “the unreconstructed α -Al₂O₃(0001) surface is Inhomogeneous and rough” discusses the stability of unreconstructed and reconstructed surface of α -Al₂O₃. The nc-AFM and DFT calculations were combined to illuminate the surface geometry. This work is extension research of the same group’s previous work published on science. The major conclusion is that the unreconstructed (1 × 1) lattice is only a metastable state, which would transfer to a reconstructed surface under high temperature. However, this statement has already been proposed by their previous work and several other articles. The authors failed to give a clear explanation for the generation of nanometer-scale (1 × 1) lattice. According to the novelty and completeness, this work does not fulfill the standard of publishing on Nature Communications. I recommend to transfer the article to other journals such as Scientific Reports.

Author response: We thank the Reviewer for the feedback and the opportunity to clarify the novelty and scope of this study. While our previous publication established the atomic structure of the thermodynamically stable ($\sqrt{31} \times \sqrt{31}$) $R_{\pm 9^\circ}$ reconstruction, it did not address the structure of the unreconstructed surface. In practice, however, the unreconstructed Al₂O₃(0001) may be of even greater relevance due to its widespread use as a substrate in catalysis and thin-film growth. The present manuscript focuses specifically on the widely assumed Al-terminated (1 × 1) surface, which has served for decades as the standard structure model in theoretical studies and as a reference substrate in experiments. The key finding of this work is that the unreconstructed Al₂O₃(0001) surface is intrinsically rough and laterally inhomogeneous, and that the Al-terminated (1 × 1) structure exists only as nanometer-scale minority domains rather than as a homogeneous termination. To our knowledge, this has not been previously demonstrated experimentally with atomic resolution. This result challenges the conventional assumption of a uniform (1 × 1) termination and has direct implications for the interpretation of prior adsorption and growth studies. The present study therefore does not merely extend our previous work, but addresses a distinct and equally important question: the true structure of the unreconstructed Al₂O₃(0001) surface under practically relevant conditions. By combining nc-AFM with DFT, diffraction, and spectroscopic data, we provide a direct experimental basis for resolving long-standing inconsistencies in the literature. We have expanded the Introduction and Discussion sections of the revised manuscript to clarify the scope of our work.

Comments on the manuscript

1. The authors argued that Al cations appeared as bright (repulsive) with a Cu tip. The same sites appeared as dark (attractive) with a CuOx tip. The brightness is not the only clue for the direction of interaction force. The $\Delta f \sim Z$ curve can be implemented to find the nature of the interaction.

Author response: We thank the Reviewer for this important comment. We agree that $\Delta f(z)$ spectroscopy can provide information about the nature of the tip–sample interaction. In the revised manuscript, we address the interaction mechanism through height-dependent nc-AFM imaging combined with corresponding probe-particle simulations. As now included in Supplementary Fig. 5, the contrast remains qualitatively unchanged over the accessible range of tip–sample distances. At closer approach, tip–surface interactions lead to imaging instabilities before entering a regime dominated by Pauli repulsion. This behavior indicates that the images in Fig. 2 were acquired in a regime dominated by electrostatic forces rather than short-range repulsion. Furthermore, we have included DFT-calculated maps of the electrostatic potential and the vertical electric field above the surface (Supplementary Fig. 6), which reproduce the

experimentally observed contrast. These calculations support the interpretation that the nc-AFM image contrast arises predominantly from electrostatic forces between the surface Al cations and the tip. We have clarified this point in the revised manuscript.

2. The mechanism of generating metastable (1×1) lattice is not well studied in the manuscript. The concentration of oxygen does not affect the surface geometry since the reconstructed phase is thermodynamically stable with a large range of oxygen chemical potentials. Discussion is required for the appearance of the unreconstructed structure. Is this structure relative to defect, step or stress? In addition, the measured lattice parameters should be given for the images in Figure 2.

Author response: As the Reviewer correctly notes, the thermodynamic stability of the $(\sqrt{31} \times \sqrt{31})R \pm 9^\circ$ is largely independent of the oxygen chemical potential. The unreconstructed surface is therefore not stabilized by oxygen stoichiometry but represents a kinetically trapped metastable state. Although the Al-terminated (1×1) configuration satisfies the requirement of surface non-polarity, it can transform to the more stable reconstructed phase once sufficient atomic mobility becomes available at elevated temperature. The existence of the (1×1) can thus be understood in terms of kinetic limitations. Although the strong inward relaxation of surface Al atoms introduces local compressive stress, the associated energy cost is small (≈ 0.1 eV per unit cell) compared to the substantial energy gain achieved upon reconstruction. We do not claim to identify the microscopic formation pathway of the unreconstructed surface. Rather, the aim of this study is to experimentally demonstrate that it deviates from the commonly assumed homogeneous (1×1) termination. The periodicity of the locally ordered domains ($a \approx 4.76$ Å) is consistent with the bulk (1×1) lattice constant. The experimental uncertainty (up to 0.25 Å) due to the small island size and lattice distortions near the edges precludes reliable quantitative determination of compressive stress from the nc-AFM images. We have added a note about the measured lattice parameters to the figure caption of Fig. 2 in the revised manuscript.

3. The authors think the unreconstructed surface is rough. This expression is not accurate since the so-called unreconstructed surface in Figure 2a is not a pure lattice. It contains multiple nanoscale lattices and other reconstructed structures.

Author response: We thank the Reviewer for this opportunity to clarify this point. By “rough”, we refer to the pronounced height corrugation and the absence of an extended, atomically flat periodic lattice on the unreconstructed surface. As shown in Fig. 2a, the unreconstructed surface exhibits height variations of more than 0.6 nm, extending over several atomic step heights within a 30 nm field of view. In contrast, the reconstructed surface shows a height variation of less than 0.1 nm over the same image size. We agree that the unreconstructed surface is not a uniform lattice. Rather, it consists of nanometer-scale (1×1) domains within a laterally inhomogeneous and corrugated morphology.

4. The thermodynamic stability of the unreconstructed lattice should be studied. Simulating a larger supercell with small structural perturbations can verify the stability.

Author response: We thank the Reviewer for this suggestion. We have explored the stability of the unreconstructed surface using larger supercells and by introducing structural perturbations, including step-like motifs (a representative example is shown in Supplementary Fig. 12). These tests did not yield any lower-energy periodic structures within the explored supercells. We emphasize, however, that the experimentally observed unreconstructed surface exhibits various step orientations surrounding confined

(1 × 1) patches and lacks long-range periodic order. Such an irregular morphology is inherently difficult to capture within conventional periodic DFT calculations, which impose translational symmetry and artificial long-range order. As a result, the experimentally observed morphology cannot be adequately represented in a finite periodic supercell. Since the complete mapping of metastable structures at the DFT level is computationally very challenging, we consider the representative model included in Supplementary Fig. 12 to provide a reasonable model for the qualitative aspects of the unreconstructed surface, as a compromise between computational feasibility and physical relevance.

5. The proportion of elements is different between two types of surfaces. The calculation for the surface energy should be very careful especially when oxygen is involved. Some necessary detail is not shown in the computational methods part. Is the zero-point energy included in the calculation? Does the calculation setup gives correct energy difference between the singlet and triplet states of O₂ molecule.

Author response: We thank the Reviewer for the thoughtful comments regarding the thermodynamic calculations. To compare surfaces with different stoichiometries, we employed the formalism of ab-initio atomistic thermodynamics (doi: 10.1103/PhysRevB.65.035406), in which the surface free energy is expressed as a function of the oxygen chemical potential, μ_{O} . This approach provides a consistent framework for evaluating the relative stability of surfaces with differing compositions. Zero-point energy (ZPE) and vibrational entropy contributions were not explicitly calculated for the surface slabs. However, when comparing structurally similar terminations of the same material, vibrational contributions to relative surface free energies are known to largely cancel. For oxide surfaces, differences in ZPE contributions between related terminations are typically only a few meV/Å² (doi: 10.1103/PhysRevB.65.035406). In contrast, the surface energy differences obtained in this work are on the order of several tens of meV/Å². Inclusion of ZPE corrections would therefore not alter the qualitative stability ordering. The reference for $\mu_{\text{O}}(T,p)$ was taken as one-half of the total energy of an isolated O₂ molecule calculated in a spin-polarized setup (VASP setting ISPIN = 2). The calculation converged to a magnetic moment of 2.0 μ_{B} , consistent with the triplet ground state of molecular oxygen. Since the thermodynamic reference requires only the ground-state energy, the singlet excited state was not explicitly evaluated. We employed the meta-GGA r²SCAN functional, which has been shown to significantly improve the predicted heats of formation for oxides (doi: 10.1021/acsmaterialsau.2c00059) and to provide a more accurate description of O₂ binding energies and magnetic states compared to standard GGA functionals (doi: 10.1021/acs.jpcclett.0c02405). We therefore consider the triplet O₂ reference energy to be sufficiently reliable for the thermodynamic framework used here. Moreover, the calculated energy of the isolated O₂ molecule affects only comparisons between structures with different stoichiometries; it does not influence relative energies of stoichiometric structures and has only a negligible effect on the comparison with the near-stoichiometric ($\sqrt{31} \times \sqrt{31}$)R±9° reconstruction variants. The relevant computational details have been clarified in the revised manuscript.

Response to Reviewer 5

The authors present a combined noncontact atomic force microscopy and density functional theory study elucidating the atomic structure of the α -Al₂O₃(0001) surface. They reveal a rough and disordered morphology, with only nanometer-scale regions exhibiting the ordered Al-terminated (1×1) structure. These results provide valuable insights for the surface science community, particularly given the widespread use of α -Al₂O₃(0001) as a substrate for thin-film growth and as a support in heterogeneous catalysis.

While the study is of clear interest, several important issues should be addressed to strengthen its robustness and impact:

1. Thermodynamic treatment of coexisting surface domains.

In Fig. 2a, it appears that approximately 50% of the surface is covered by (1×1) domains, with the remainder being disordered regions. Although Fig. 1d indicates that the (1×1) surface is less stable than the reconstructed surface upon high-temperature annealing, the calculations appear to consider only a pure (1×1) phase. Given the experimental observation of coexisting ordered and disordered domains, their coexistence may alter the thermodynamic landscape and potentially stabilize the surface relative to a fully reconstructed phase. A more rigorous treatment would ideally consider mixed ordered/disordered surface configurations, even though modeling disorder is technically challenging.

Author response: We thank the Reviewer for the positive assessment of our work and for recognizing its relevance. We agree that the experimentally observed unreconstructed surface is predominantly disordered, with only a small fraction exhibiting the (1 × 1) termination. The surface energy comparison shown in Fig. 1d refers to the idealized periodic surface terminations and establishes the strong thermodynamic preference of the reconstructed phase over a flat, homogeneous (1 × 1) surface. We agree that the surface energy of the experimentally observed coexistence of ordered and disordered regions cannot be determined with our DFT calculations. Such a mixed configuration represents a metastable, spatially heterogeneous structure rather than a well-defined phase. Capturing this coexistence rigorously would require large, non-periodic models with explicit disorder, which are not tractable within conventional periodic DFT and would introduce arbitrariness in selecting representative configurations. To illustrate the energetic effect of roughening, we have included a model that captures some qualitative aspects of the disordered surface (see Supplementary Fig. 12). Although this model does not reproduce the full experimental disorder, it demonstrates that introducing locally stepped motifs is energetically inexpensive relative to the ideal flat (1 × 1) termination.

2. Nature and origin of the disordered domains.

What is the atomic nature of the disordered regions? Do they contain both Al and O species, or are they dominated by one? Given their large surface coverage (~50%), there must be a thermodynamic driving force for their formation. A clearer microscopic picture of the mechanisms responsible for forming and stabilizing these disordered domains would significantly strengthen the interpretation of the experimental observations.

Author response: We thank the Reviewer for this important question. Based on our experimental and computational results, the disordered regions can be consistently interpreted as predominantly oxygen-terminated configurations. The overall composition is very close to stoichiometric Al₂O₃, as indicated by

XPS (Supplementary Figs. 7 and 8), consistent with the expectation that non-stoichiometry (leading to a polar surface) would be energetically highly unfavorable in a wide-bandgap material. Constant-height nc-AFM is primarily sensitive to the outermost atomic layer and does not directly resolve the atomic arrangement in the lower-lying regions surrounding the Al-terminated (1×1) patches. The best indication of the qualitative properties of the disordered surface currently available is the stepped surface model in Supplementary Fig. 12. This model contains motifs in which surface Al atoms adopt tetrahedral or octahedral coordination similar to that found in the reconstructed surface, rather than the metastable threefold planar coordination characteristic of the ideal (1×1) termination. A predominantly oxygen-terminated surface is also consistent with the observed low chemical reactivity reported in the literature, whereas an extended Al-terminated surface would be expected to be significantly more reactive. We have clarified this interpretation in the Results section as well as in the figure captions of Supplementary Figs. 7, 8, and 12 of the revised manuscript.

3. Annealing conditions in Supplementary Fig. 2.

What is the annealing environment in Supplementary Fig. 2: vacuum or O₂? If O₂ is used, what is the pressure? How do these conditions differ from the air annealing reported in Supplementary Fig. 1?

Author response: All annealing steps shown in Supplementary Fig. 2 were performed in 1×10^{-6} mbar O₂, except for the final annealing step at 1300 °C, which was carried out in UHV. However, the oxygen partial pressure during annealing did not affect the resulting LEED patterns. High-temperature annealing led to the formation of the $(\sqrt{31} \times \sqrt{31})R \pm 9^\circ$ reconstruction under both O₂ and UHV conditions. The sample shown in Supplementary Fig. 1 was annealed in a tube furnace in air, and the large-scale morphology was evaluated by ambient AFM measurements performed in air. We have clarified the annealing conditions in the revised manuscript.

4. Annealing conditions in Supplementary Fig. 5.

Similarly, please clarify the annealing environment in Supplementary Fig. 5 (vacuum vs O₂), including the O₂ pressure if applicable, and how it compares with the air annealing conditions in Supplementary Fig. 1.

Author response: For both datasets shown in Supplementary Fig. 5 (now Supplementary Fig. 8), the sample was annealed in 1×10^{-6} mbar O₂ dosed into the UHV chamber: at 900 °C to prepare the unreconstructed surface and at 1300 °C to obtain the $(\sqrt{31} \times \sqrt{31})R \pm 9^\circ$ reconstruction. In Supplementary Fig. 1, the sample was annealed in a tube furnace in ambient air. We have added the annealing conditions to the revised figure caption.

5. Role of O₂ in Supplementary Fig. 6.

In Supplementary Fig. 6, annealing is performed in 1×10^{-6} mbar O₂. Why was O₂ used instead of vacuum, as in Supplementary Fig. 2? Is O₂ required to drive the transition from (1×1) to the reconstructed surface, and if so, why? Is there any difference in the transition temperature or pathway between O₂ and UHV annealing? If the surface does not lose lattice oxygen during annealing, the role of O₂ in this process should be clarified.

Addressing these points would significantly enhance the mechanistic understanding and general impact of the work.

Author response: High-temperature annealing in either UHV or in 1×10^{-6} mbar O_2 leads to the formation of the $(\sqrt{31} \times \sqrt{31})R_{\pm 9^\circ}$ reconstruction. The oxygen partial pressure does not affect the transition temperature or final surface structure, consistent with our result that both structures are stoichiometric. Varying the O_2 pressure at fixed temperature produced identical LEED patterns. O_2 was introduced during annealing primarily to facilitate the removal of carbonaceous contaminants from samples transferred from ambient atmosphere and was retained in subsequent experiments for consistency. The reconstruction is thermodynamically favored over the entire range of practically attainable oxygen chemical potentials, as reflected in the calculated phase diagram (Fig. 1d), and consistent with previous studies reporting formation of the reconstruction even at substantially higher O_2 pressures (doi: 10.1002/adma.202312899). Moreover, our XPS data (Supplementary Fig. 8) do not indicate any loss of lattice oxygen during annealing. The transition is therefore governed primarily by atomic mobility at elevated temperature rather than by changes in oxygen stoichiometry. We have clarified this point in the revised manuscript.

Response to Reviewers of

The Unreconstructed α -Al₂O₃(0001) Surface is Inhomogeneous and Rough

Johanna I. Hütner-Reisch, Andrea Conti, David Kugler, Florian Mittendorfer, Michael Schmid, Ulrike Diebold, Jan Balajka

Review round: 2

We thank the Reviewers for their positive evaluation of the revised manuscript and for their constructive suggestions. We have addressed these points in the current revision and provided responses to the remaining questions below.

Response to Reviewer 2

The manuscript by Hütner-Reisch et al. was evaluated by 5 reviewers and the revisions made by the authors considerably improved it. Still, I found that my major concern that the argumentation how the authors establish chemical selectivity with two different tip terminations, is not sufficiently addressed. The authors argue that electrostatic interactions govern the contrast, but as said before, setting the one tip positive and the other negative in the simulations basically determines the outcome. Therefore, the choice of positive/negative tip charge has to be justified in more detail. Ref. 42. provides not sufficient support to assume a negative tip charge for CuO-tip and the positive charge at the metallic apex is also just assumed. Due to the importance of this issue for the main conclusions of the paper, a more substantial argumentation is needed. I recommend considering the following papers to do so: (1) [10.1088/1367-2630/ab8efd](https://doi.org/10.1088/1367-2630/ab8efd) (2) [10.1021/acsnano.4c03155](https://doi.org/10.1021/acsnano.4c03155) (3) [10.1021/acs.nanolett.5b05251](https://doi.org/10.1021/acs.nanolett.5b05251)

Author response: We thank the Reviewer for this important comment regarding the interpretation of the nc-AFM contrast and the assignment of the surface species. We agree that the choice of tip charge must be justified and that the identification of the observed features should not rely on assumed parameters. In the revised manuscript, we now support the assignment of a negatively charged CuOx tip and a positively charged Cu tip by incorporating the suggested literature (DOI: [10.1021/acsnano.4c03155](https://doi.org/10.1021/acsnano.4c03155), [10.1088/1367-2630/ab8efd](https://doi.org/10.1088/1367-2630/ab8efd), and [10.1021/acs.nanolett.5b05251](https://doi.org/10.1021/acs.nanolett.5b05251)).

Response to Reviewer 4

The manuscript is greatly improved after revision. Although I am not fully convinced that “the Al-terminated (1 x 1) surface cannot be experimentally obtained”, this manuscript still reveals new fact to the terminating surface of Al₂O₃. I would recommend to publish this article on Nature communications.

If possible, the authors can give some effort on further improvement of this article regarding the following issues.

Author response: We thank the Reviewer for the positive assessment of the revised manuscript. Regarding our statement that “the Al-terminated (1 × 1) surface (...) cannot be experimentally obtained”, we agree that we cannot provide a strict proof that it is impossible to create such a surface. We therefore change the sentence to “...is not observed in experiment.”

1. The authors use a reconstructed surface model with steps to argue that large-scale (1 x 1) structure is not stable. But the results show that steps are not energetically favored compared to the bare surface. The readers would expect some disordered structures (vacancies, adatoms, stacking faults, steps, et al) giving lower formation energy than the surface.

Author response: We agree that a realistic model of the inhomogeneous surface could provide a lower-energy configuration than the ideal (1 × 1) termination. However, modeling the experimentally observed highly irregular surface is intractable within currently available ab-initio methods, even when aided by machine-learned force fields. The main challenge lies in the large number of possible supercells, for each of which the number of Al₂O₃ units must be varied in a wide range to describe different widths and depths of the trenches around the (1 × 1) area(s). We have explored two different supercells with varying numbers of Al₂O₃ units (24 configurations in total; for each of them 10 parallel tempering runs with 12 images each). These calculations did not yield any structures with a surface energy lower than that of the model in Supplementary Fig. 12. This is not an exhaustive search, and we now explicitly state this limitation in the Discussion. Nevertheless, we are confident that the model shown in Supplementary Fig. 12 captures many qualitative aspects of the unreconstructed surface, in particular the tendency toward oxygen termination and increased coordination of surface Al atoms.

The experimentally observed rough morphology does not necessarily imply that its surface energy is lower than that of the ideal (1 × 1) termination. The roughness may also arise from the mass transport required for dehydration of a surface previously exposed to the ambient environment (and may persist up to annealing temperatures of ≈1000 °C due to the low energetic cost of step formation). To clarify this point, we add a new Supplementary figure (Supplementary Fig. 13) illustrating why dehydration requires mass transport, which would lead to a stepped surface.

In addition, the XPS spectra shows reduced Al-O ratio. This ratio should also be provided in their (1 x 1) ordered and disordered structures.

Author response: In the Supplementary Information of the revised manuscript, we provide [O]/[Al] ratios for both experimentally accessible surfaces: the unreconstructed (disordered) surface and the (√31 × √31)R±9 reconstructed (ordered) surface.

- Unreconstructed surface: 1.52 (normal emission, 0°), 2.41 (grazing emission, 70°)
- Reconstructed surface: 1.59 (normal emission, 0°), 2.19 (grazing emission, 70°)

The higher [O]/[Al] ratio at the more surface-sensitive 70° emission angle indicates that both surfaces are predominantly oxygen-terminated, in contrast to the (1×1) Al-termination. Moreover, the comparable [O]/[Al] ratios for these two surfaces at normal emission indicate similar overall stoichiometry, close to Al_2O_3 .

2. The authors switched the position of panel g and panel c-f in figure 2. I can not find the meaning of this treatment. On line 139, The sentence “Above $\approx 1000^\circ\text{C}$, the surface transformed into an atomically flat and well-ordered $(\sqrt{31} \times \sqrt{31})\text{R}\pm 9^\circ$ reconstruction (Fig. 2b).” The authors changed Fig. 2g to Fig. 2b in the revised manuscript. This seems to be a typo.

Author response: We thank the Reviewer for pointing out this error. In the previous revision, we rearranged Fig. 2 so that panel (g) is directly below panel (a) for easier visual comparison, as suggested by Reviewer 3. The figure reference in the text has now been corrected.